# New Eocene primate from Myanmar shares dental characters with African Eocene crown anthropoids

Jean-Jacques Jaeger[1], Olivier Chavasseau [1], Vincent Lazzari[1], Aung Naing Soe[2], Chit Sein [3], Anne Le Maître [1,4], Hla Shwe[5] & Yaowalak Chaimanee[1]

Recent discoveries of older and phylogenetically more primitive basal anthropoids in China and Myanmar, the eosimiiforms, support the hypothesis that Asia was the place of origins of anthropoids, rather than Africa. Similar taxa of eosimiiforms have been discovered in the late middle Eocene of Myanmar and North Africa, reflecting a colonization event that occurred during the middle Eocene. However, these eosimiiforms were probably not the closest ancestors of the African crown anthropoids. Here we describe a new primate from the middle Eocene of Myanmar that documents a new clade of Asian anthropoids. It possesses several dental characters found only among the African crown anthropoids and their nearest relatives, indicating that several of these characters have appeared within Asian clades before being recorded in Africa. This reinforces the hypothesis that the African colonization of anthropoids was the result of several dispersal events, and that it involved more derived taxa than eosimiiforms.

[1] Laboratory PALEVOPRIM, UMR CNRS 7262, University of Poitiers, 6 rue Michel Brunet Cedex 9, 86073 Poitiers, France. [2] University of Distance Education, Mandalay 05023, Myanmar. [3] Ministry of Education, Department of Higher Education, Naypyitaw 15011, Myanmar. [4] Department of Theoretical Biology, University of Vienna, Althanstrasse 14, 1090 Vienna, Austria. [5] Department of Archaeology and National Museum, Mandalay Branch, Ministry of Religious Affairs and Culture, Mandalay 05011, Myanmar. Correspondence and requests for materials should be addressed to J.-J.J. (email: jean-jacques.jaeger@univ-poitiers.fr)

The evolutionary history of anthropoids, in particular their geographical origin, has been intensely debated during the last decades[1–5]. Earlier hypotheses advocating an African origin for anthropoids have been overturned by the discovery of older and phylogenetically more basal anthropoids in Asia, which are thought to comprise the clade Eosimiiformes[1,3,6–8]. Recent findings in Libya and Myanmar[3,9] have shown that the eosimiiforms dispersed from Asia to Africa during the Middle Eocene. Anthropoids are also represented in the Southeast Asian Eocene by another group, the amphipithecids[10–14], the affinities of which remain a matter of debate. However, crown anthropoids and their nearest fossil relatives do not appear to be closely related to these eosimiiforms, leaving their ancestors unknown. They appear successively in the African fossil record (proteopithecids, parapithecoids, oligopithecids, and propliopithecids) between the late middle Eocene and the earliest Oligocene. Available data converge to support multiple dispersal events from Asia to Africa by anthropoid taxa, which were more derived than eosimiiforms.

A new primate was collected from the Pondaung Formation, about 2 km thick continental formation of Central Myanmar from which only lower half of the Upper Member has yielded fossil mammals[15]. These fossiliferous deposits have been recently radiometrically redated from 40.31 to 40.22 Ma (late Middle Eocene)[16] and are therefore older than any known African anthropoid locality. The new fossils were found in Paukkaung Kyitchaung 2 (PK2) locality, nearby the village of Paukkaung (Myaing township, Magway Region), a rich mammalian locality that has previously yielded several remains of other anthropoid primates such as eosimiiforms, amphipithecines, and also siva-ladapid strepsirrhines[3,12,17–19]. They have been excavated from a fluviatile clayish and sandy layer and were associated with diverse terrestrial and aquatic vertebrate remains, including fishes, crocodiles, and turtles[20]. The new primate from the PK2 locality of Pondaung Formation (Myanmar) does not belong to either of the two Asian anthropoid clades, the eosimiiforms[1,3,6–8] and the amphipithecids[10–14], but shares instead derived dental characters with some groups of the Eocene crown anthropoids and their nearest fossil relatives. It therefore may bridge the gap between Asian and African anthropoids.

## Results

**Systematic paleontology**. Order Primates Linnaeus, 1758
    Suborder Anthropoidea Mivart, 1864
    Family incertae sedis
    *Aseanpithecus* gen. nov.
    *Aseanpithecus myanmarensis* sp. nov.
**Etymology** ASEAN, after the political and economic Union of Southeast Asian countries and "*pithecus*" meaning monkey in Latin. The species name refers to the country where the fossil was discovered.
**Holotype** NMMP 93 (Fig. 1a, b and Supplementary Fig. 1) left maxilla with anterior orbital rim and C-M$^3$ lacking M$^1$ and buccal part of P$^4$. Holotype and hypodigm fossils are housed in the Paleontology Collections of the National Museum, Ministry of Culture in Naypyitaw, Myanmar.
**Hypodigm** NMMP 95 (Fig. 1c–g and Supplementary Fig. 1) anterior fragment of a right lower jaw with partial canine alveolus, P$_2$–P$_3$, and P$_4$ alveoli and M$_1$ mesial root, and NMMP 96 (Fig. 1h, i and Supplementary Fig. 1) right M$_3$.
**Type locality** PK2 locality, Paukkaung village, Myaing Township, Central Myanmar. The holotype and the hypodigms have been collected in very close proximity.
**Age** Late Middle Eocene Pondaung Formation.
**Measurements** (see Table 1)

**Diagnosis** Medium-sized anthropoid (similar in tooth size to *Ateles* with estimated body weight between 2.8 and 3.4 kg) (Supplementary Note 1) characterized by its short and elevated muzzle, small and forwardly located orbits with low convergence (Supplementary Note 2), large upper canine with mesial groove, three premolars with reduced unicuspid P$^2$, P$^3$, and P$^4$ unwaisted, buccolingually broad with a small, mesially located protocone, no lingual cingulum, and reduced styles and buccal cingula. Subrectangular M$^2$ with large trigon basin, peripheralized and low cusps, U-shaped protocristae, complete and low crests without paraconule but with a tiny swelling, which may correspond to a vestigial metaconule, reduced buccal cingulum, strong and continuous mesial, lingual and distal cingulum without hypocone or pericone, and weak hypoparacrista. Upper molars trigone basins with slight enamel crenulations. Deep lower jaw with most probably unfused symphysis, large canine alveolus, P$_2$-P$_3$ with strong protoconids surrounded by complete cingulids, stronger lingually than buccally, and with small paraconids and hypoconids. P$_2$ with drop-like occlusal outline, massive single root with crown surface similar to that of P$_3$. P$_2$ larger than P$^2$. P$_3$ with stronger hypoconid, without talonid basin nor metaconid. M$_3$ with very short trigonid basin and large and deep talonid basin, with a large, central hypoconulid. Entoconid fused into entocristid, closing the talonid lingually. Main cusps bunodont with rounded wear facets.

It differs from the eosimiiforms by its larger size, larger P$^2$, P$^3$ with a more extended lingual lobe and a distinct protocone, upper molars with small styles and buccal cingula, less distinct hypoparacrista, more bunodont and peripheralized cusps, subrectangular M$^2$ outline. It differs from the amphipithecines by its lower premolar structure and proportions and by the absence of a distolingual cusp on upper molars. It differs from the proteopithecids by its less convergent orbits, P$^3$–P$^4$ with less developed protocones and lacking incipient hypocone, upper molars with more bunodont cusps, less developed buccal cingula, lacking hypocone and paraconule, weaker hypoparacrista, and stronger lower canine. It differs from parapithecoids by its shorter muzzle, less convergent orbits, less bunodont cusps on upper molars without additional enamel cusplets, unicuspid P$^2$, P$^3$–P$^4$ length/width proportions, and the absence of hypocone on P$^3$–M$^3$, P$_2$ as large as P$_3$. It differs from the oligopithecids and the propliopithecids by its less convergent orbits, the retention of P$^2$/$_2$, absence of hypocone and hypoparacrista, P$^3$-P$^4$ more buccolingually extended, P$^3$ with a more mesially located paracone and a sharper and longer postparacrista, with smaller protocone and no lingual cingulum, simple bulbous P$_3$ with no honing facet and stronger lingual cingulid, nonconcave distal margin of their upper molars (except *Catopithecus*). It differs from platyrrhines by its less convergent orbits, its smaller P$^2$ and more bunodont cusps on upper molars. It differs from *Perupithecus* by its larger molar size, more rectangular outline, larger trigon basin, weaker hypoparacrista, more bunodont cusps, absence of metastyle, and more lingually located parastyle.

**Description**. NMMP 93 (Fig. 1 and Supplementary Fig. 1): The left maxilla is nearly complete. It displays the alveolar region with C-M$^3$ (buccal part of P$^4$ and M$^1$ are missing), the frontal ascending process and the anterior rim of the orbit. It shows a unique maxillary component of the ventral orbital rim between the jugal and the upper margin of the ascending process. Lacrimal bone, foramen, and the associated canal are not present on the smooth external surface and must have been therefore located inside the orbits, a characteristic anthropoid character[21]. In lateral view, its outline indicates an abbreviated and elevated muzzle characterized by strong canine crown and root as in *Bahinia*[8].

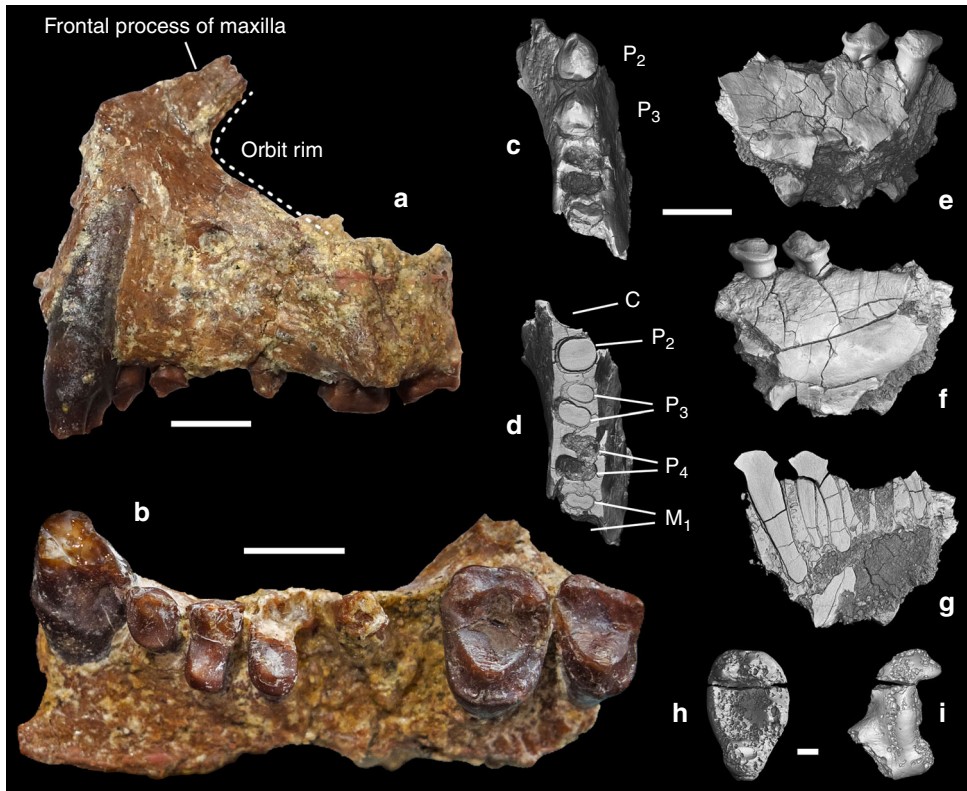

**Fig. 1** *Aseanpithecus myanmarensis* gen. et sp. nov. **a**, **b** NMMP 93 holotype left maxilla with C-M$^3$, lacking M$^1$ and buccal part of P$^4$. **a** Buccal view. **b** Occlusal view. **c–g** NMMP 95 right mandible with canine alveolus, P$_2$–P$_3$, P$_4$ alveoli and M$_1$ mesial root. **c** Occlusal view. **d** Virtual horizontal section at the root level. **e** Buccal view. **f** lingual view. **g** Virtual parasagittal section of the mandible showing the long root P$_2$. **h–i** NMMP 96 right M$_3$. **h** Occlusal view. **i** Lingual view. Scale bars for **a**, **b**, and **c–g** 5 mm. Scale bar for **h–i** 1 mm

| Table 1 Dental measurements of *Aseanpithecus myanmarensis* gen. et sp. nov | | | |
|---|---|---|---|
| **Specimen number** | **Tooth** | **Mesiodistal length** | **Buccolingual width** |
| NMMP 93 | C | 4.03 | 4.16 |
| | P$^2$ | 2.19 | 2.42 |
| | P$^3$ | 2.42 | 3.69 |
| | P$^4$ | (4.09) | (4.46) |
| | M$^1$ | – | – |
| | M$^2$ | 4.71 | 5.85 |
| | M$^3$ | 4.0 | 5.04 |
| | P$^2$-P$^4$ | 8.46 | |
| | C-M$^3$ | 25.38 | |
| NMMP 95 | P$_2$ | 3.43 | 2.93 |
| | P$_3$ | 3.34 | 3.07 |
| NMMP 96 | M$_3$ | 6.13 | 3.99 |
| All measurements are in millimeters. Measurements in parenthesis are estimated values | | | |

The orbit is small (estimated diameter of 13.4 mm obtained in adapting the methodology of reference[22] to the preserved portion of the orbit rim) and proportionally small relative to tooth size, indicating diurnal activity (see later). Frontation can be only estimated by comparisons but not quantified, due to the incompleteness of the orbit. However, the frontation appears to be higher than that of basal haplorrhines, such as *Teilhardina asiatica*[23] but lower than in *Proteopithecus*[24], *Parapithecus grangeri*[25], and *Apidium*[26].

Concerning orbital convergence, the absence of midline reference point and of orbital superior point make it impossible to orient the specimen reliably enough to measure precisely convergence following the methodology of reference[27]. Nevertheless, we have estimated an orientation of the orbit plane based on three points selected on the preserved anterior rim of the orbit. We have also produced a digital symmetric replica of the left maxilla in order to associate the two maxilla to define a putative midline. From this reconstruction, we have estimated a minimum value for convergence of 35°. The anterior margin of the orbit is located above the line between P$^4$ and M$^1$ as in *P. grangeri*[25] being therefore less anteriorly located than in *Bahinia*[8] and in *Proteopithecus*[24]. Canine and canine jugum are strongly developed, with a long root, inducing a well-marked canine fossa. The maxilla is only slightly longer (25.38 mm) than high (20.04 mm)

and its suborbital part displays high elevation (13.30 mm), a distinctive anthropoid character[7,28]. The maxillary mesial border is very elevated, only slightly oblique, running parallel to the canine mesial root border, like in *Bahinia* and to a lesser degree in *P. grangeri* and *Apidium*. Shortly above the tip of canine root, the maxillary bone bends distally into the frontal ascending process with an angle of 150° as in *P. grangeri*. The lower orbital rim is preserved only in its mesial part and is broken away in its distal part, from the level of distal part of $M^1$. There is also no contact between the zygomatic and lacrimal bones along the ventral orbital region, unlike in omomyids, tarsiids, and strepsirrhines[29]. The root of the broad zygomatic arch, which is broken away, lines up above the $M^2$ in an anteriorly oriented oblique direction as in most anthropoids. The zygomatic is not in contact with an external lacrimal bone as observed in strepsirrhines. A rounded and single infraorbital foramen is well developed and situated 4.98 mm below the mesial extremity of the orbit. The frontal ascending process of the maxilla is rather wide and strongly inclined, becoming progressively narrower. Its mesial rim follows the nasal bone, which is missing, as is the premaxilla. The maxillary sinus is weakly developed. In palatal view, the muzzle appears as broad, the anterior palatine foramen is not distinguishable, and was therefore located farther anteriorly than the canine. The palate is wide mesially at the level of the canine, and then becomes narrower until $P^3$–$P^4$ contact level, increasing in width distally, and reaching its maximal width at the level of $M^2$ and zygomatic arch root. The maxillary alveolar region contains a canine, three premolars, and two molars. The first molar and the buccal part of $P^4$ are missing; only one mesiobuccal root of $M^1$ being preserved. The canine is very large, in root diameter and crown elevation. In cross section, its outline is roughly circular. Its apex is missing. A deep mesial groove extends from the broken apex to the root. Two other shallower grooves are located lingually and distobuccally, respectively. The canine crown distal part displays a sharp crest, which descends from the apex. A cingulum surrounds the crown and intersects this distal crest. On its lingual side, the cingulum surrounds a marked depression made by $P_2$. Three premolars are present, rather small in proportion to the molars, their mesiodistal length representing only 30% of the C-$M^3$ length. $P^2$ is a small, single-rooted, unicuspid tooth with a nearly rounded, oval outline. Its transverse diameter is only slightly larger than its mesiodistal one. There is no lingual expansion or cusp but a cingulum surrounds completely its crown, being more strongly expressed on the lingual basal part of the crown. Two small swellings, located on cingulum, correspond to para- and metastyles, respectively. $P^3$, which is larger than $P^2$, displays a mesiodistal narrow crown without waisting and is buccolingually expanded. Its lingual lobe is rather wide mesiodistally relative to its buccal lobe since it represents 65% of buccal crown length. It displays a small protocone, which is mesially bent and situated on the mesial side of the lingual lobe. From the paracone, a strong postparacrista crest joins a reduced metastyle. Pre-paracrista is short and steep. A weak distal cingulum connects the metastyle to the protocone, and there is no other connecting crest between paracone and protocone. A very shallow buccal cingulum is present, but no trace of lingual cingulum occurs. $P^4$ buccal part is missing but its outline can be inferred. It is larger than $P^3$ and its lingual lobe is also elongated buccolingually. Its lingual lobe is larger than that of $P^3$ with a similarly mesially bent small protocone. The first molar is only represented by its mesiobuccal root but according to the space available between $P^4$ and $M^2$, its size should have been slightly larger than that of $M^2$. $M^2$ is large, buccolingually broad, and displays a subrectangular outline, its mesiodistal diameter being only slightly greater buccally than lingually. Its trigon is large, with only three main cusps connected by distinct, but low,

crests. No paraconule or hypoparacrista is present but a tiny swelling corresponding to a metaconule is visible. The protocone is the main and highest cusp and is located centrally. It is peripheralized, being close to the lingual wall. Two distinct crests, pre-protocrista and postprotocrista join respectively a small parastyle and the metacone. Paracone is slightly higher than metacone and the two cusps are peripheralized, rather small, low, and rounded with weakly developed postparacrista and pre-metacrista. A shallow buccal cingulum is present. The parastyle is low and located slightly more lingually than the paracone. The distal crest, which is less elevated than the mesial one, arcs distobuccally towards the metacone, isolating a large trigon basin. A strong and continuous cingulum circles the mesial, lingual, and distal walls of the tooth, with many enamel wrinkles. No hypocone or pericone are present. $M^3$ resembles $M^2$ in its structure. It is not strongly reduced in size. Its metacone is only slightly smaller and lower than the paracone and its lingual lobe is narrower than that of $M^2$. Its buccal wall is only slightly more oblique than that of $M^2$ and its main cusps are also peripheralized. The buccal cingulum is very weakly developed, as on $M^2$. Its distal wall is slightly indented, unlike $M^2$.

NMMP 95: This right lower jaw has been collected less than one meter away from the maxilla. It matches in size and the characters displayed by both the lower jaw and the maxilla do not match any other fossil primate recovered for 20 years in this Formation. Its teeth share several characters that are also represented on the maxilla, such as similar size, strong vertical canines, high elevation of the horizontal ramus of the mandible, and robust anterior dentition. It consists of a horizontal branch fragment with the distal part of canine alveolus, complete $P_2$, $P_3$, and $P_4$, and mesial $M_1$ alveoli. It displays an incomplete but rather deep horizontal ramus (more than 10.24 mm under $P_3$). The symphysis is broken but a ridge of its planum alveolare is preserved, which slopes down with an angle of about 45° (Supplementary Fig. 1j). Its possible lowermost extremity has been preserved, located at the level of $P_2$ crown and combined with the angle made by the planum alveolare it suggests that incisors roots were not strongly horizontally inclined. In occlusal view, the lingual wall of the jaw starts to curve lingually at the level of the distal part of $P_3$. The partial lingual alveolus indicates the large size of the slightly inclined canine. The unicuspid $P_2$ has an oval, drop-like occlusal outline with a narrower mesial part. Its apex is broken. The tooth displays a similar occlusal surface than $P_3$, as in platyrrhines, *Serapia* and *Proteopithecus* in which $P_2$ is even larger than $P_3$. Its single root is slightly inclined mesially, very strong and long, being even longer than those of $P_3$. Its crown is basally inflated and its mesial part projects mesially. A complete cingulid, more strongly developed lingually than buccally, surrounds the base of the crown. A paraconid cusplet is present on this cingulid, and is connected to the protoconid by a distinct preprotocristid. A tiny hypoconid occurs on the distal cingulid. $P_3$ is similar in outline and structure to $P_2$, but displays a weaker paraconid and a larger and higher hypoconid. There is no distinct talonid basin. A short and very shallow distolingual protocristid departs from the protoconid apex but no metaconid swelling is visible at its end. $P_4$ is only represented by its two alveoli and must have been larger than $P_3$. A micro-computed tomography (micro-CT) virtual horizontal section shows that $P_3$–$P_4$ roots are oriented slightly obliquely along the long axis of the tooth row as in most other contemporary anthropoids.

NMMP 96: A lower right $M_3$ was found from wet screening of the sediments, which have also yielded NMMP 95 and the holotype maxilla. Its enamel is heavily corroded due to enamel dissolution by terminal plant rootlets. Judging from its size and morphology (bunodont cusps, large talonid basin, and peripheralized cusps), it corresponds well to what is expected to

characterize the $M_3$ of this new taxon and is therefore tentatively referred to it. It is characterized by a maximal transverse diameter at the hypoconid level, bunodont cusps, large talonid, and mesiodistally short trigonid. The protoconid is larger but lower than the metaconid. It is slanted buccally. The trigonid basin is open mesiolingually. The hypoconid is large, in a mesial position and connected to a large but narrow hypoconulid lobe. There is a slightly oblique cristid obliqua. The entoconid is reduced, located on an uninterrupted crest closing lingually the talonid. The talonid is large buccolingually and mesiodistally with a worn and rounded hypoconulid.

These three new fossils, recovered by wet screening, originate from the same place, only a few decimeters away from each other. Being of similar size and tooth wear level, they could belong to the same individual. However, the fluviatile sands of this site (PK2) never have delivered so far associated fossils of a single individual. Therefore, we consider as unlikely that these fossils belong to the same individual.

*Aseanpithecus* lived in sympatry with amphipithecids (*Pondaungia* (*Amphipithecus*), *Ganlea* (*Myanmarpithecus*), eosimiiforms (*Afrasia*, *Bahinia*), and sivaladapids (*Kyitchaungia*, *Paukkaungia*)). Comparisons with these primates show that *Aseanpithecus* displays a clearly original mosaic of primitive and derived characters that testifies to its belonging to a distinct group of Asian anthropoids.

Most of the *Aseanpithecus*-derived characters are shared with anthropoids. The short and high rostrum, the high elevation of the maxilla between the lower orbital margin and the alveolar level, the exclusive presence of the maxilla along the mesial and ventral orbital margin, the absence of lacrimal foramen and canal in front of the orbit and subsequently the absence of contact between the jugal and the lacrimal bones, the high elevation of the horizontal ramus of the lower jaw, the obliquely oriented alveoli of the roots of $P_3$ and $P_4$, the single-rooted $P^2/_2$, the $P_2$ nearly as large than $P_3$ and larger than $P^2$, are the characters whose association characterizes the anthropoids. Some of these features can be found isolated among strepsirrhines[30] and are thus homoplasic, like the mesially grooved upper canine or the single-rooted $P_2$[31], but no known strepsirrhine displays as many anthropoid characters as *Aseanpithecus*. Many strepsirrhines display rather narrow lower premolars with root alveoli in line with the long axis of the tooth row, two-rooted $P^2/_2$, a lacrimal foramen and canal in front of the orbit, a contact between jugal and lacrimal bones along the ventral orbital margin, low elevation of maxilla below the orbit, long rostrum, shallow mandibular corpus, more oblique symphysis, less bunodont cusps, and stronger buccal shearing crests on upper molars. Some authors have proposed that the amphipithecids are anthropoid-like strepsirrhines, which have developed their anthropoid characters as a result of their hard-object feeding habits[32]. *Aseanpithecus* shares several of these alleged skull and dental characters, such as strong canines, short and high rostrum, deep mandibular corpus, or bunodont molars[33]. These features are associated with a dentition indicative of the consumption of insects/fruits rather than hard objects. In addition, several of these features are present in the primitive eosimiiform anthropoid *Bahinia*, which was also not a hard-object feeder. Therefore, at least for *Aseanpithecus*, and perhaps also for all amphipithecids, these shared characters can be instead interpreted as those of primitive Asian anthropoids.

**Comparison with *Bahinia* (eosimiiform).** *Bahinia*[8] shares with *Aseanpithecus* a similar maxilla structure (great anterior depth, nearly vertical premaxilla-maxilla suture, high suborbital elevation). Both share also a strong vertical canine root and a small orbit. However, the frontal process of *Aseanpithecus* seems to have been less vertically oriented than that of *Bahinia*. Their dentitions differ by several important characters. *Bahinia* $P^{2/}_2$ are significantly smaller than $P^{3/}_3$. $P_2$ reaches the same size as $P_3$ in *Aseanpithecus* and its buccal cingulids are also more strongly developed. $P^3$ of *Bahinia* is less buccolingually elongated and lacks protocone. Its length/breadth ratio is 77% against 66% for *Aseanpithecus*. Molars of *Bahinia* differ by their strong buccal cingula and styles, smaller trigone basins because of more lingually slanted paracones and metacones, and the presence of a distinct hypoparacrista.

**Comparisons with amphipithecids.** The Pondaung amphipithecines differ by many anatomical traits from *Aseanpithecus*. Their maxillary anatomy is poorly documented but still indicates a lower elevation below the orbits[18]. Their palate shape is different in occlusal view, displaying a parabolic outline. Moreover, their teeth are quite distinct. $P_2$ and $P_3$ are strongly expanded distolingually and $P_2$ is more reduced compared with $P_3$ than in *Aseanpithecus*. Their upper premolars have stronger and more medially located lingual cusps, which are connected to the paracone by two crests. Upper molars and $P^4$ of amphipithecines have wrinkled enamel, variably developed conules and always show a distolingual cusp variously interpreted as a true hypocone[28], a pseudo-hypocone[34], or a displaced metaconule[35]. *Siamopithecus*[14], sometimes considered as a stem amphipithecid, differs by the structure of its upper molars, yielding a hypocone connected to the protocone by a strong prehypocrista and slanted trigone cusps. Its upper premolars, wide mesiodistally, are more derived with strong lingual cusps on $P^3$-$P^4$. $P^2$ is reduced. Lower premolars of *Siamopithecus* differ by the strongly reduced $P_2$ and the more buccolingually expanded $P_3$ with a more concave lingual wall and a more convex buccal wall and which displays also weaker cingulids. *Siamopithecus* $M_3$ is also elongated, with a short trigonid, strong hypoconid and hypoconulid, and a reduced entoconid situated on an uninterrupted lingual crest closing the talonid basin. *Bugtipithecus* is another basal amphipithecid[36] sometimes assigned to an even more basal position in anthropoid phylogeny[5,10,23,37]. Its upper premolar displays a narrower lingual lobe with a stronger protocone and molars show less peripheralized main cusps, a larger metaconule, a deeper trigone basin, a strong hypocone connected by prehypocrista to postprotocrista, strong parastyles, and a stronger buccal cingulum. Its $M_3$ has a longer trigonid, a more distally located hypoconid and a notch interrupting the entocristid. *Aseanpithecus* dentition shares therefore mostly primitive characters with eosimiids and amphipithecids and lacks their autapomorphic characters. On the contrary, it has developed distinct derived characters some of which are found among the oldest noneosimiiform African and South American anthropoids.

**Comparison with *Altiatlasius koulchii*.** This enigmatic Paleocene primate from Morocco[38] has upper molars with very bunodont cusps, strong metastyles, short pre- and post-protocristae, two conules, reduced trigone basin. Protocone is higher than the buccal cusps. $M^3$ has a strong parastyle, reduced metacone, and weak pre- and post-protocristae. No shared derived characters with *Aseanpithecus* can be pinpointed and its phylogenetic position remains obscure.

**Comparisons with African and South American anthropoids.** *Aseanpithecus* shares several dental characters with *Proteopithecus*, *Serapia,* and platyrrhines ($P_2$ as large as or larger than $P_3$ associated to a reduced $P^2$, strong and mesially grooved upper canine) but differs from them by its more primitive upper

premolars with smaller lingual cusps. In addition, *Proteopithecus* molar structure is different (high trigone cusps, strong hypocone and buccal cingulum, incomplete lingual cingulum). Concerning the molars, the best morphological match is found with the oligopithecid *Catopithecus*. Both share upper molars with a large and shallow trigon basin, U-shaped pre-and postprotocristae, reduced buccal cingulum, rectangular outline of the lingual lobe, complete and strong lingual cingulum, and absence of indentation of the distal wall. However, *Catopithecus* differs by the presence of a small hypocone, a hypoparacrista and by its less bunodont cusps and higher trigone crests. Similar molar structure is observed in *Oligopithecus* in which a hypocone is not always present and the buccal cingulum is weak. *Oligopithecus* differs also from *Aseanpithecus* by the marked concavity of the distal wall of its $M^2$—but not for its $M^3$ which is also concave in *Aseanpithecus*—and the development of the hypoparacrista. However, the anterior dentition of oligopithecids is quite distinct and much more derived, displaying only two premolars and an enlarged $P_3$ with a honing facet. The oldest-known oligopithecid *Talahpithecus*[9] and the oldest-known South-American primate *Perupithecus*[39] also display a similar upper molar structure. Both differ from *Aseanpithecus* by a less peripheralized protocone and therefore by a smaller trigon basin, the presence of incipient hypocone (or putative hypocone for *Talahpithecus*) and pericone, well-developed hypoparacrista and distal wall indentation.

The teeth of the parapithecoids display little resemblance with those of the new fossil. The most primitive parapithecoid, *Biretia*, has low, bunodont premolars and molars with inflated cusps, developed conules, and reduced/absent connecting crests between trigone cusps on upper molars. Its $P_2$ is smaller than its $P_3$. Finally, it shares only the small single-rooted $P^2$ with *Aseanpithecus*[40].

$M^2$ of *Aseanpithecus* shows resemblance with the upper molar of *Perupithecus* (see above). Other resemblances with platyrrhines include the proportions between $P^2$ and $P_2$, the equivalent size of $P_2$ and $P_3$ (usually $P_2$ is larger than $P_3$ in platyrrhines) which are also found among *Branisella*[41]. The dentition in its whole displays also some resemblance with that of the Miocene callithricine *Lagonimico*[42]. We note shared characters in the large canine with an anterior groove extending to the root, premolar structure (narrow and buccolingually elongated with narrow lingual lobe and small protocone), upper molar structure (absence of conules and hypocone, large trigon basin, strong and complete lingual cingulum, reduction of buccal cingula, low rounded molar shearing crests). However, differences are also numerous (*Aseanpithecus* possesses a different shape of the dental arcade, less reduced posterior molars, absence of metaconid on $P_3$, absence of lingual lobe on $P^2$, more reduced lingual cingulum on $P^4$, larger upper molar trigons). Some of these characters, like the absence of hypocone, are considered to be secondarily lost by these callitrichines[42]. If so, the resemblance with *Aseanpithecus* would reflect shared feeding adaptations rather than phylogenetic affinities. More fossils are necessary to understand these characters and definitely polarize them.

**Comparison with *Amamria*.** *Amamria* is represented by a complete upper molar, which has been described from a 39.5 Ma Tunisian locality[43]. This tooth is considered to represent a transitional form between eosimiiforms and derived African anthropoids. However, this tooth appears to be closer morphologically to strepsirrhines than to eosimiiforms. If isolated molars of eosimiiforms resemble those of primitive strepsirrhines, the occurrence of a distinct paraconule, the weakness of hypoparacrista, the low elevation and incompleteness of the postprotocrista and hypo-metacrista, the occurrence of a distinct

hypocone and an incipient pericone are rarely associated in primitive anthropoids and tilt the balance rather to the side of strepsirrhines. The morphology of *Amamria* does not support any close relationship with the much more derived *Aseanpithecus*.

**Phylogenetic position**. The combination of a short rostrum, the great depth of the maxilla under the orbit, the significant contribution of the maxillary bone to the ventral orbital region and the limited participation of the jugal to the lower orbit rim, the absence of an external lacrimal foramen, the large and subvertical canine with a deep mesial groove, the smaller size of $P^2$ relative to $P_2$, the broad unwaisted upper premolars with protocone on $P^3$-$P^4$, the upper molar structure, the single-rooted $P_2$, the oblique orientation of $P_3$-$P_4$ roots relative to the long axis of the tooth row, characterize *Aseanpithecus*, and testify to its anthropoid affinities. Its orbit was estimated to have a diameter of 13.4 mm. Using an estimated length of 5 mm for $M^1$, this orbit diameter is proportionally small relative to tooth size, indicating a diurnal activity pattern (Supplementary Fig. 3). Most of the available dental characters correspond to those hypothesized for the archetype of crown anthropoids[2]. Two distinct clades of Paleogene anthropoids are known from Asia, the amphipithecids and the eosimiiforms. The amphipithecines are characterized by an association of uniquely derived characters, including the distolingual expansion of their anterior lower premolars, reduced $P_2$, and constant presence of an additional distolingual cusp on their upper molars. None of them shares derived characters with *Aseanpithecus*. The same is true for the basal amphipithecid *Siamopithecus*. The eosimiiforms are considered as the sister group of all other anthropoids[3,5,44] and *Aseanpithecus* shares with them several of their plesiomorphic characters: reduced single-rooted $P^2$ without protocone, large canines, crestiform upper molars without distinct conules or hypocone, $P_3$ without metaconid. Its derived characters exclude it from this basal clade, but some resemblance occur, concerning plesiomorphic anthropoid characters. According to the fact that *Bahinia* represents the most derived eosimiid known and that eosimiids are considered as the stem group of all later anthropoids, such a resemblance of *Aseanpithecus* to its ancestors is expected and normal.

The maximum parsimony phylogenetic analyses performed based on a dataset of 324 morphological characters and 45 taxa, all retrieve *Aseanpithecus* as an anthropoid. Moreover, they do not support *Aseanpithecus* as pertaining to the eosimiiforms, this taxon being invariably nested within a clade encompassing all derived anthropoids (Amphipithecidae + Parapithecoidea + Proteopithecidae + Oligopithecidae + Propliopithecidae + Platyrrhini). In our first phylogenetic analysis (heuristic search with full taxonomic sampling, some multistate characters ordered and no topological constraint enforced) two equally parsimonious trees of 1497 steps were found (Fig. 2). In the obtained strict consensus tree, *Aseanpithecus* is nested within a clade of derived anthropoids composed of the Amphipithecidae, a Parapithecidae + Proteopithecidae clade, the Oligopithecidae, the Propliopithecidae, and the Platyrrhini. *Aseanpithecus* is grouped with *Bugtipithecus* in a clade that is reconstructed as the sister group of the crown anthropoids, here composed of the platyrrhines, the propliopithecids, and the amphipithecids. The most basal representatives of the derived anthropoids on this consensus tree are the Parapithecidae + Proteopithecoidea clade and the oligopithecids. Seven additional analyses have been performed to test the sensibility of the phylogenetic position of *Aseanpithecus* (see the "Methods" section). Depending on the settings of the analysis, the retrieved phylogenetic position of *Aseanpithecus* within the Amphipithecidae + Parapithecoidea + Proteopithecidae + Oligopithecidae + Propliopithecoidea + Platyrrhini clade is variable,

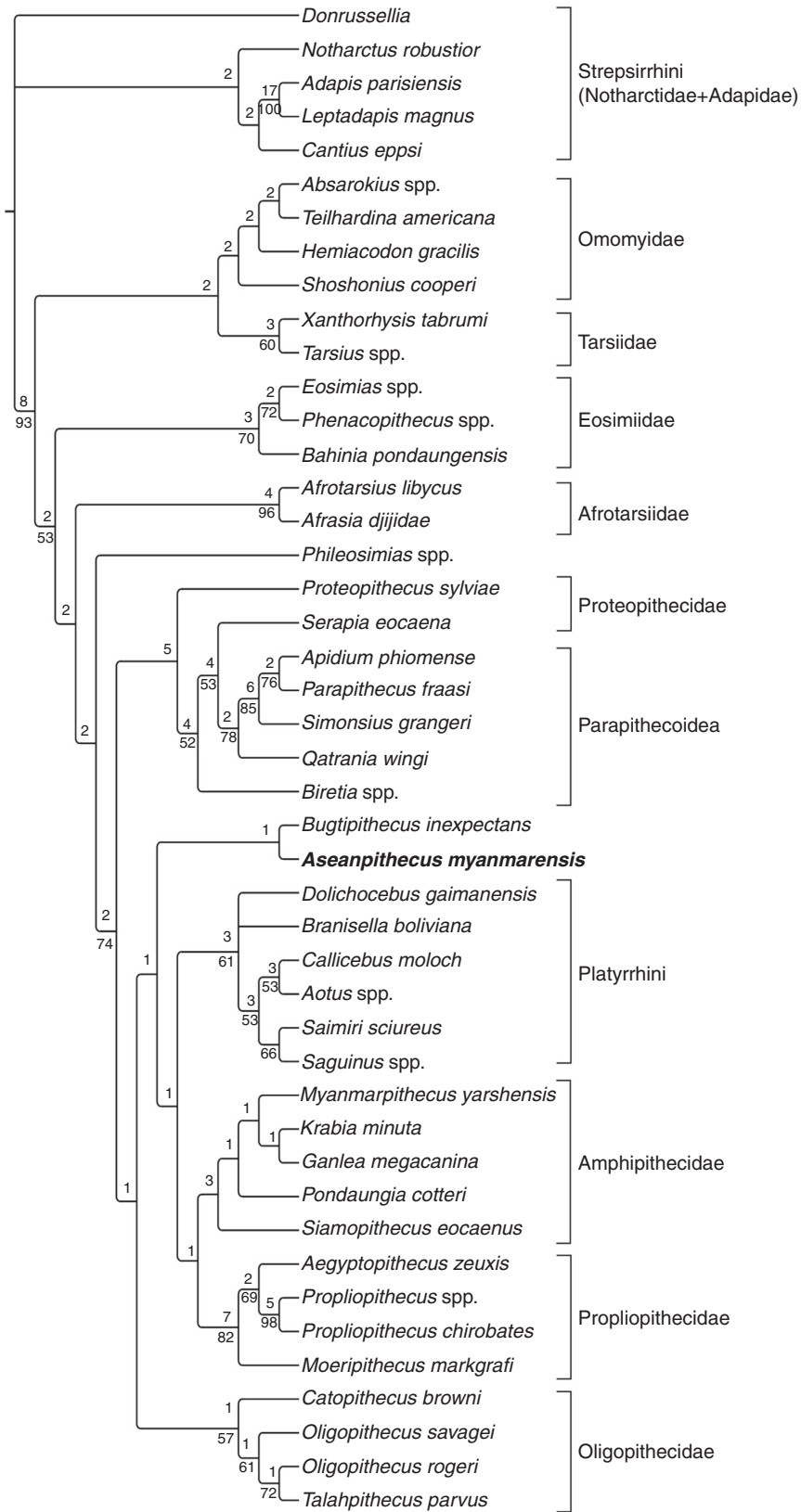

**Fig. 2** A possible phylogenetic position of *Aseanpithecus myanmarensis* gen. et sp. nov. Maximum parsimony analysis performed with PAUP 4b10 with a datamatrix of 45 taxa and 324 morphological characters (heuristic search with some multistate characters treated as ordered, no topological constraint enforced, all characters equally weighted). Strict consensus of two equally parsimonious trees of 1497 steps. Consistency index (CI) = 0.3220, Homoplasy index (HI) = 0.6780, CI excluding uninformative characters = 0.3010, HI excluding uninformative characters = 0.6990, Retention index (RI) = 0.5579, and Rescaled consistency index (RC) = 0.1796. Bremer support values are indicated above the nodes. Bootstrap support values (>50%) are indicated below the nodes

likely reflecting its original combination of primitive and derived features. It is either placed basally (two of eight analyses) or, more frequently, deeply nested within this clade (six of eight analyses) and often phylogenetically close to the clade of crown anthropoids (Supplementary Note 3 and Supplementary Figs. 4–7). Because of the apparent instability of *Aseanpithecus* in the obtained trees, reflected by low Bremer and bootstrap support values of several internal nodes within the clade of the derived anthropoids, our phylogenetic analysis does not support the attribution of this genus to any known family of anthropoids and rather suggests that it may belong to a new family.

## Discussion

*Aseanpithecus* displays a unique combination of derived premolar and molar characters associated to plesiomorphic orbit features, confirming the hypothesis that the cranial characters evolved more slowly than the dental characters in anthropoids[29]. *Aseanpithecus*, the oldest-known African oligopithecid *Talahpithecus*[9] and the oldest South American anthropoid *Perupithecus*[39] share a similar upper molar structure (peripheralized molar cusps, absence of conules, U-shaped protocristae, and strong and united mesial, lingual, and distal cingula) likely corresponding to the ancestral upper molar morphotype of crown anthropoids[2] plus *Aseanpithecus*. However, the upper molars of *Aseanpithecus* differ from the others by the weakness of their hypoparacrista, the complete absence of hypocone and pericone, by their straight distal wall and by their more bunodont and lower cusps. However, the morphological variation of *Aseanpithecus* upper molars is known from only one specimen, preventing more detailed interpretations. Nevertheless, it suggests that the accentuated bunodonty of the propliopithecids and the parapithecoids likely represents a derived state[2] (*contra* references[45,46]).

According to its original mosaic of primitive and derived characters, *Aseanpithecus* likely documents a new family of Asian anthropoids, which is dentally significantly more derived than eosimiiforms and therefore more closely related to crown anthropoids. The primitive characters relate it to the eosimiiforms, which are considered as the stem group of all others anthropoids. The derived characters of *Aseanpithecus* are different from those found among the endemic Southeast Asian family Amphipithecidae, but are shared with several groups of African and South American anthropoids. Upper molars are similar to those of *Catopithecus*, a derived oligopithecid and premolars (at least $P^3$ and $P^4$), display the same organization, but at a less derived level. Other derived characters, such as the $P_2$ larger or equivalent in size to $P_3$, and reduced $P^2$ associated with large $P_2$, are shared with other groups, like proteopithecids and platyrrhines. *Aseanpithecus* cannot be considered as a close relative of these African/South American taxa but rather as a member of a new Asian clade of anthropoids that has the potential to develop the original characters of these African/South American groups. An alternative hypothesis would be that the derived features of *Aseanpithecus* have been acquired convergently compared with crown anthropoids. However, the absence of a clear African ancestor to crown anthropoids and the important number of shared derived characters make us favor the previous interpretation.

It is also crucial to better constrain the timing and nature of the "out-of-Asia" dispersal of *Aseanpithecus* relatives. Eosimiiforms have been recently considered to have colonized Africa and, being recorded in Asia since the middle Eocene[6,7] (circa 45 Ma) and in Africa as early as 39 Ma[3,9], their dispersal must have occurred before 39 Ma. Several dispersal scenarios have been proposed for anthropoids out of Asia to Africa. The most parsimonious one consists of a unique colonization event by an eosimiiform,

followed by a rapid adaptive radiation leading to most or all African clades[41,44]. Nevertheless, the discovery of *Aseanpithecus* suggests instead that the Eocene colonization of Africa by Asian anthropoids may have involved several dispersal events and distinct groups, some of them having been more derived than eosimiiforms[3] and yielding derived characters also found among the taxa of the African radiation. Other mammalian groups of Asian origins, such as hystricognath, anomaluroid rodents[47–49], and anthracothere cetartiodactyls[50], also dispersed from Asia to Africa during Eocene. But if some hystricognath rodents seem to have been associated to the earliest anthropoid dispersal wave[9], the anthracotheres are not documented in Africa before the Latest Eocene[51] and may testify to later dispersal events from Asia to Africa, which could have included anthropoids.

In conclusion, *Aseanpithecus* can be interpreted as a pivotal fossil primate documenting a transitional stage between primitive Asian anthropoids and African crown anthropoids and their nearest fossil relatives. Thus, a more diversified initial adaptive radiation of Eocene Asian anthropoids, that included the ancestors of some, if not all, African clades, has probably occurred in Asia. Accordingly, one can forecast the discovery in the future of additional anthropoid clades in the Asian Eocene faunas.

## Methods

**Micro-computed tomography scan imaging.** The lower jaw, isolated M3, and the cast of the maxilla remains were scanned using an EasyTom HR-microtomograph (Platform PLATINA of IC2MP, University of Poitiers) with voxel sizes of 20 μm (surface scan of the maxilla), 8.95 μm (mandible) and 9.2 μm (M3). Scan parameters for the mandible: X-ray voltage = 60 kV, current = 38 μA, number of projections = 2880, filter = Tukey, framerate = 4 frames s⁻¹. Scan parameters for the M3: X-ray voltage = 60 kV, current = 37 μA, number of projections = 992, filter = Tukey, framerate = 4 frames s⁻¹. Virtual 3D models were extracted from the micro-CT image stacks to compute virtual slices imaging the alveoli and dental roots of the fossils. Three-dimensional surface PDF files were also computed from the virtual models using the Geomagic (3D Systems) software (Fig. 1). The original upper maxilla could not be micro-CT scanned, the Myanmar primate fossils, other than those found in screening residue, being not authorized to be exported.

**Phylogenetic analysis.** Phylogenetic analysis was carried out with PAUP 4.0b10[52] using the hsearch command (heuristic search) with randomized addition of taxa (1000 replications). Three hundred and twenty four dental, cranial, and postcranial characters were coded for 45 taxa including 40 fossil taxa and 5 extant taxa. In total, eight different analyses were carried out to test the sensibility of resulting topologies to the inclusion of the unstable anthropoid *Phileosimias*, the treatment of character state (partly ordered vs. completed unordered), and the use of topological constraints extracted from recent phylogenetic analyses of anthropoids[5,53]. The complete description of taxonomic sampling, characters list, and obtained trees is included in Supplementary Note 3.

**Estimation of orbit diameter.** The diameter of the orbit was estimated using the methodology of reference[22]. This methodology is using three points of the orbit plane (X, Y, and Z) to calculate the orbit radius. The original methodology for point selection is using the inferior most point of the orbit (Y) and two other points along the orbit (X and Z) equally distant from Y[22]. The specimen was oriented following the orbital plane, photographed, and the WY and WZ distances have been subsequently determined with the software ImageJ. The formula (corrected from that of reference[22], erroneous) used to calculate the orbit diameter D was $D = 2OY = (WY^2 + WZ^2)/WY$. See also Supplementary Note 2.

**Nomenclatural acts.** This published work and the nomenclatural acts it contains have been registered in ZooBank, the proposed online registration system for the International Code of Zoological Nomenclature. The ZooBank LSIDs (Life Science Identifiers) can be resolved and the associated information viewed through any standard web browser by appending the LSID to the prefix "http://zoobank.org/". The LSIDs for this publication are: urn:lsid:zoobank.org:pub:3C15AC67-0CCE-405D-8F22-94206A2816C3; urn:lsid:zoobank.org:act:0F4A1575-A69A-491A-9FD5-F58594AF5750; urn:lsid:zoobank.org:act:5E738887-64BB-4DA7-9DD6-768AF44D08A1.

**Reporting summary.** Further information on research design is available in the Nature Research Reporting Summary linked to this article.

## Data availability

The data supporting the article are available as Supplementary Information with the exception of the datamatrix used for the phylogenetic analysis, which is available as Supplementary Data 1. Three-dimensional pdfs, 3D printable surface files, and TIF image series of *Aseanpithecus myanmarensis* have been permanently deposited in the figshare "Aseanpithecus myanmarensis" (https://doi.org/10.6084/m9.figshare.c.4560509). The following items are available in this collection: NMMP 93, holotype left maxilla: 3D pdf, 3D printable surface file in STL format; NMMP 95, right mandible: 3D pdf, 3D printable surface file in STL format, TIF image series; NMMP 96, right $M_3$: 3D printable surface file in STL format, TIF image series; 3D pdf of the symmetrized composite jaws of *Aseanpithecus myanmarensis*.

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

## Acknowledgements

We thank the villagers from Bahin and Paukkaung in Pondaung area, Myanmar and their Chairmen for their help, kindness, and enthusiasm that greatly facilitated our fieldworks, K. C. Beard (Kansas University) and S. Ducrocq (PALEVOPRIM) for providing comparative materials, S. Riffaut (PALEVOPRIM) for drawing picture, A. Mazurier (Platform PLATINA of IC2MP, University of Poitiers) for scanning of fossils, M. Rugbumrung (DMR) for preparing and casting fossils. This work has been supported by the CNRS UMR 7262, the University of Poitiers, the Ministry of Culture and Ministry of Education of Myanmar and funded by the National Geographic Society Foundation (W344-14) and the Leakey Foundation to J.-J.J., the ANR 17-CE02-0010-01 DieT-PrimE program (to V.L.) and the ANR-DFG 18-CE92-0029/BO 3479/7-1 EVEPRIMASIA program (to O.C.).

## Author contributions

J.-J.J., O.C., V.L. and Y.C. wrote the paper, O.C. and V.L. performed analyses, and J.-J.J., O.C., V.L., A.N.S., C.S., A.L.M., H.S. and Y.C. contributed to field data acquisition.

## Additional information

**Competing interests:** The authors declare no competing interests.

