## [Peer Review File · Nature Communications]

Reviewers' Comments:

Reviewer #1:

Remarks to the Author:

It was a pleasure to review the paper by Jaeger et al. entitled: New Middle Eocene primate from Pondaung formation (Myanmar) shares dental characters with African Eocene crown anthropoids. This paper presents the discovery of three new fossil specimens from the Pondaung formation in Myanmar, attributed to a new taxon *Aseanpithecus myanmarensis*. Following morphological and cladistical analysis, the authors propose this new form as a member of a new clade of Asian early anthropoids intermediate between the more primitive eosimiiformes and more derived African forms.

The paper is well structured and easy to follow. I would have no special remarks regarding its organization. The logic is solid and the conclusions leading to the proposal of a new anthropoid clade, look reasonable, despite the somehow floating position of this new taxon on the different cladistics analysis. Nevertheless, it is very difficult to judge on the validity of several key characters based on the provided iconography.

Considering the number of characters described, I have the feeling that many illustrations are missing to demonstrate their validity. Most of these additional figures could be in supplementary information, but from my point of view, they are really important to strengthen the paper and to allow other researchers to observe these characters more in details. I list hereafter in bullets style the additional illustrations I would have like to see:

- Real photographs of the three specimens, under 6 classical orthogonal views, with eventually stereo-pairs for the teeth.
- The photographic assemblage of the symmetric maxilla used for the orbit analysis described in the paper, in order to be able to judge on the pertinence of the angular measurements and other descriptions linked to general facial aspects. In the same way, to illustrate the putative midline presented by the authors, the same maxilla with symmetry in occlusal view. By the way, even if the specimen cannot leave Myanmar for μ CT scanning, other techniques would allow simple surface 3D acquisition such as laser scanner, or even photogrammetry that can be done with a simple camera and free software. Having such 3D surface would make many observations and measurement much easier. It would also allow to make a rather complete dento-gnathic reconstruction of the specimen using all the available material to discuss more in details specific morphological aspects.
- A mirror completed mandible (based on μ CT data) as well as a detail of the symphyseal region to justify the unfused and rather vertical characters of the symphysis (it is absolutely impossible on the provided picture to see if any part of the symphysis is present).
- A 3D segmentation of the dental roots in the mandible could be a nice illustration, but is not critical considering the characters observed on these structures.

In addition to these illustrations, some other aspects would require additional comments in the paper or in the supplementary information. First of all, the M3 that is tentatively attributed to this new taxon presents strong surface degradation. A short discussion about it would be desirable, especially if it could be interpreted as digestion. If it is a common preservation artifact of this site, it should be mentioned.

The drawing of the outline of the M1 looks a bit weird to me. Could it not have been a bit shifted lingually on the figure? The logic of this outlined M1 should be better explained in the text.

Finally, regarding the CT aspects, I would like to have much more details regarding the acquisition parameters, with at least the source voltage and intensity, the filters used, the number of projection and the exposure time / frame averaging. In general, these parameters should always be provided when using μ CT / CT data in order to be able to judge about the quality of the scanning. There is a mistake in the method part: the maxilla and the mandible are mentioned, when the specimens

scanned are the mandible and the M3.

Still considering μ CT data, it would be useful to have the data used for this paper publicly available, as it becomes more and more often the case. If not possible, clear statement explaining why it is not possible and under which condition the data would be accessible would be necessary. Nevertheless, increasing the number of publicly available data for such fossil specimens is the only way to really make possible in depth comparisons going beyond the surface morphology only. Making data public when using CT scanning would be really valuable for the whole profession. When slices cannot be made available for legal reasons, at least surface files should be provided especially as 3D pdfs have been apparently generated.

To conclude, this is a nice paper that I recommend for publication, providing that additional illustrations are included to demonstrate the validity of several key characters presented by the authors, and I strongly encourage to go for publicly available μ CT data.

Paul Tafforeau

Reviewer #2:

Remarks to the Author:

This new maxilla and more fragmentary mandible from Myanmar are new and interesting fossils to consider for anthropoid evolution. They add a new combination of dental characters from what is known in Asia. They will be controversial as their interpreted status as a "crown" anthropoid versus a "stem" anthropoid will be debated. Making the connection from Asia to African anthropoids is the heart of the matter in terms of an evolutionary sequence and biogeographical origins. I think that these new fossils are deserving of a publication in Nature. They will certainly provide discussion and debate. I would also point out to the authors that there is postcranial evidence from Shanghuang that indicates a more advanced anthropoid existed there with the eosimiids although no dental material has been described.

Reviewer #3:

Remarks to the Author:

Jaeger and colleagues report the discovery of two jaw fragments and an isolated tooth from the Late Middle Eocene of Myanmar. They attribute these to a new genus and species of anthropoid primate, and offer an interpretation of the significance of its morphology in the context of its age and geographical location. Their main conclusion is that this new species reveals something significant about the phylogenetic and biogeographical connections among the Paleogene anthropoids of Asia and Afro-Arabia. This conclusion is based on the perceived similarities that this taxon shares with later anthropoids from Afro-Arabia.

I see potential problems with three key premises in their argument: 1) that these specimens belong to one taxon, 2) that these specimens share derived features with later Afro-Arabian anthropoids not seen in Asian Paleogene anthropoids, and 3) that the perceived similarities have a potentially coherent interpretation given their distribution within Afro-Arabian Paleogene anthropoids. I explain these concerns below, and attach an image that compiles comparative illustrations of upper molars pertinent to the discussion from various publications. The authors have seen these, I know, but I include them so that they can see some of what I was looking at when making the observations that follow.

1) Attribution of the specimens. The new species is known from three specimens – an upper and lower jaw, and an isolated m_3 . These seem to match adequately in size, and all appear to belong to an anthropoid, so their association is understandable. However, there are other anthropoid taxa in the same beds, and the number of taxa has grown over the years. So it is not unrealistic to think that there is more than one previously undescribed species from this locality, rather than just one. After all, years ago, there were two primates there, then three, then four... This is always a potential problem when establishing new taxa, and I only raise it here because the upper and lower jaws seem to resemble very different sets of taxa, as will be made clear below. This makes me worry that the two jaw fragments belong to different taxa.

2) The apomorphies of *Aseanpithecus*. It is argued in this manuscript that *Aseanpithecus* exhibits a set of derived morphologies not seen in other Paleogene Asian anthropoids. In the upper molar(s) these are chiefly the reduced buccal cingula, parastyles, metastyles, paraconules and metaconules, enlarged trigon basin, and reduced concavity of the distal molar margin. I believe that these are either seen, or are easily produced by wearing the known molars of *Bahinia*, *Phileosimias*, and *Bugtipithecus*. The upper molars show fairly heavy wear, despite the apparent lack of exposed dentin. The buccal surface of the protocone is clearly very worn. This wear would, because of the geometry of the protocone, preprotocrista, and postprotocrista, have the effect of enlarging the basin. The morphology seen here could easily be achieved by wearing down the lingual side of a molar that started out with a more restricted basin, like those of *Bahinia pondaungensis* or *Phileosimias*. That such wear was occurring in this dentition is clearly corroborated by the wear on the premolars.

The buccal portion of the molar crowns do lack the pronounced styles and shelf-like cingulum often seen in eosimiids, but these features are poorly developed in some taxa – like *Bahinia pondaungensis*, *Bugtipithecus*, and *Phileosimias*. These taxa also lack the pronounced concavity of the distal crown margin. At this point, it is worth noting that much of this could also be said of *Altiatlasius* from the Paleocene of Africa. The lack of any reference to this genus in the paper is a major shortcoming.

The upper premolars of *Aseanpithecus* are, in my view, not demonstrably different from those of *Bahinia*. The morphology of *Bahinia*'s premolars has been difficult to discern with certainty because the original illustration in Jaeger (1999) inaccurately reconstructs them. Comparison of more recent imagery of the specimen shows that the P³ and P⁴ are incomplete buccally and heavily worn lingually. This has several consequences. First, the relative buccolingual dimensions of the two teeth cannot be precisely determined. The P³ seems narrower, but the incomplete nature of both teeth, and the potential buccal displacement of the P³ crown, prevent us from falsifying the possibility that the two are very similar in breadth (as in *Aseanpithecus*). Second, the assertion that *Bahinia* lacks a P³ protocone is difficult to confirm in the type specimen. This similarity to *Bahinia* combined with the near lack of direct evidence for the premolar morphology of *Phileosimias* and *Bugtipithecus* leaves open the possibility that there is little difference among them. I realize that the P³ assigned to *Bahinia banyueae* has no protocone, but the generic assignment of that species could be debated.

In sum, the known upper teeth of *Aseanpithecus* do not seem very different from those of several known Asian Paleogene anthropoids. It is entirely possible that it is closely related, if not congeneric, to one of them.

In the lower dentition, the main apomorphy is the relative size of the P₂ and P₃, with the P₂ being at least as large a crown (though single-rooted) as the P₃. This feature of the lower premolars is indeed unlike the size sequence in *Eosimias*, *Myanmarpithecus*, and *Pondaungia*, but *Bugtipithecus*, *Phileosimias* and *Siamopithecus* are not known from specimens that can demonstrate the size sequence of the premolars, and *Bahinia pondaungensis* is difficult to interpret with certainty (the known mandibular specimen has a P₃ that is damaged in such a way that it appears to be larger than P₄, which is surely an artifact). These taxa, in which the premolar size sequence is ambiguous, include taxa that I've argued resemble *Aseanpithecus* in the morphology of its upper dentition. This being the case, it is again possible that *Aseanpithecus* (assuming that its upper and lower jaws belong together) is a close relative of something like *Phileosimias*, *Bugtipithecus*, or *Bahinia*.

3) Distribution of *Aseanpithecus*'s apparent apomorphies among Afro-Arabian anthropoids. This last issue may seem the most nebulous of the three, but in fact it may be the most problematic because it threatens the validity of their argument, while the first two problems concern only the veracity of individual premises.

Although, as argued above, some of the purported apomorphies of *Aseanpithecus* are found in other Asian taxa, some have so far only been documented in Afro-Arabian fossils. However, their distribution in Afro-Arabian taxa is odd. The upper molar features are most closely matched by the stem catarrhine *Catopithecus* (in which the premolars are profoundly different), while the lower premolar traits are found in proteopithecids (in which the upper molars are completely different). So the lower teeth resemble proteopithecids (stem anthropoids), and the upper molar (but not premolars) resembles a stem catarrhine – and one that is typically

considered more derived than *Oligopithecus*. Parenthetically, some of these molar traits are also present in the Paleocene *Altiatlasius*, which complicates their polarity.

This pattern of distribution could support a premise like “*Aseanpithecus* exhibits features not found in other Asian Paleogene anthropoids, but found in various combinations in later Afro-Arabian Paleogene anthropoids”. As explained above, I’m not sure that even this is accurate because some of these traits are found in Asian taxa like *Phileosimias* and even in the *earlier* African *Altiatlasius*. So even this vague premise is generous. But my present point is that because these traits do not align *Aseanpithecus* with any particular Afro-Arabian clade, there is no particular paleobiogeographical scenario to which this argument can lead. It is tantalizing, and the specimens may well end up being a meaningful part of the solution, but at present they offer only an intriguing clue. They are suggestive of something, but what? I wouldn’t argue that such riddles can only be explored or solved by using massive cladistic data sets, but in such complicated situations, it is a good starting point. The authors have conducted such an analysis, and the unstable position of *Aseanpithecus* under various assumption sets reflects the ambiguity just described, but it also clearly shows its propensity for clustering with *Bugtipithecus*. If one tries (as does the parsimony algorithm) to hypothesize a way in which this new species connects more primitive Asian Paleogene anthropoids with later and more derived Afro-Arabian ones, one might imagine a dispersal of an *Aseanpithecus*-like group into Africa, where it is somehow involved in the evolution of oligopithecids or proteopithecids. But where do they fit?

Are they somewhere between proteopithecids and *Catopithecus*? Do they show us the initial changes toward the catarrhine lineage? If this is the case, why does the upper molar resemble *Catopithecus* instead of *Oligopithecus*? And why would it exhibit a large P₂, when these are absent in oligopithecines? Did the lineage enlarge them before losing them?

Or is it a stem anthropoid like a proteopithecid. If so, its similarities to *Catopithecus* are all convergent, and its only potential synapomorphy with anything in Africa is the large P₂— which we can’t be sure is not present in some Asian taxa (or even *Altiatlasius*).

Even if I set aside my concerns about the composition of the hypodigm, and the overstated differences between *Aseanpithecus* and all known Asian taxa, and concede that the upper teeth look just like *Catopithecus*, and the lower teeth like proteopithecids, it still leaves us with no way of interpreting these similarities without one set being convergent and irrelevant. Honestly, I’d sooner believe that these fossils document two taxa - an early oligopithecid, *and* an early proteopithecid. I know it would be a more sensational claim, but it would follow more clearly from the available evidence. In fact, I’d encourage the authors to uncouple the upper and lower dentitions of *Aseanpithecus* and try them experimentally as separate OTUs. I’d be curious to see where the analysis puts them.

Lastly, I want to make clear that it is entirely possible that the authors’ conclusions are right. I am pointing out what I see as problems with their argument. If these weaknesses cannot be remedied, then their argument is the worse for it, but hopefully my criticisms can improve their argument. As always, I am skeptical of the argument, not of the authors.

Very low relief of the protocone, pre- and postprotocristae caused by heavy wear

Remains of preparaconule crista. Paraconule erased by wear.

Heavy wear

Reviewer #1 (Remarks to the Author):

It was a pleasure to review the paper by Jaeger et al. entitled: New Middle Eocene primate from Pondaung formation (Myanmar) shares dental characters with African Eocene crown anthropoids. This paper presents the discovery of three new fossil specimens from the Pondaung formation in Myanmar, attributed to a new taxon *Aseanpithecus myanmarensis*. Following morphological and cladistical analysis, the authors propose this new form as a member of a new clade of Asian early anthropoids intermediate between the more primitive eosimiiformes and more derived African forms.

The paper is well structured and easy to follow. I would have no special remarks regarding its organization. The logic is solid and the conclusions leading to the proposal of a new anthropoid clade, look reasonable, despite the somehow floating position of this new taxon on the different cladistics analysis. Nevertheless, it is very difficult to judge on the validity of several key characters based on the provided iconography.

Considering the number of characters described, I have the feeling that many illustrations are missing to demonstrate their validity. Most of these additional figures could be in supplementary information, but from my point of view, they are really important to strengthen the paper and to allow other researchers to observe these characters more in details. I list hereafter in bullets style the additional illustrations I would have like to see:

- Real photographs of the three specimens, under 6 classical orthogonal views, with eventually stereo-pairs for the teeth.

We could not provide real pictures under 6 views of the specimens because these specimens are presently stored in Myanmar and thus unavailable for new photographs. In addition, additional photographs would not significantly improve the presently provided illustrations of these specimens that we consider as appropriate for readers.

We have improved and enriched the illustrations of the fossils by providing stereo-pair images of the teeth in occlusal view. These stereo-pair images have been realized with CT-scan images considering that the specimens are presently unavailable for acquisition of new photographs.

- The photographic assemblage of the symmetric maxilla used for the orbit analysis described in the paper, in order to be able to judge on the pertinence of the angular measurements and other descriptions linked to general facial aspects. In the same way, to illustrate the putative midline presented by the authors, the same maxilla with symmetry in occlusal view. By the way, even if the specimen cannot leave Myanmar for μ CT scanning, other techniques would allow simple surface 3D acquisition such as laser scanner, or even photogrammetry that can be done with a simple camera and free software. Having such 3D surface would make many observations and measurement much easier. It would also allow to make a rather complete dento-gnathic reconstruction of the specimen using all the available material to discuss more in details specific morphological aspects.

We have provided a symmetrized version of the maxilla in frontal and occlusal views (now available in Supplementary figure 1)

- A mirror completed mandible (based on μ CT data) as well as a detail of the symphyseal region to justify the unfused and rather vertical characters of the symphysis (it is absolutely impossible on the provided picture to see if any part of the symphysis is present).

We have provided a symmetrized version of the mandible (now available in Supplementary figure 1).

Concerning the remarks of reviewer 1 about the symphyseal region of the mandible. Even if the sagittal plane is not preserved on the specimen, parts of the planum alveolare and the distal end of the symphysis are preserved in a near midline section (see additional picture provided). These features document a much vertical symphysis.

Annotated symphyseal region of NMMP 95 showing preserved parts of the planum alveolare and distal termination of the symphysis.

- A 3D segmentation of the dental roots in the mandible could be a nice illustration, but is not critical considering the characters observed on these structures.

We have not considered that it was necessary to provide a 3D rendering of the mandible with roots visible in transparency. We think that this view would not bring more morphological information than the virtual longitudinal section already provided.

In addition to these illustrations, some other aspects would require additional comments in the paper or in the supplementary information. First of all, the M3 that is tentatively attributed to this new taxon presents strong surface degradation. A short discussion about it would be desirable, especially if it could be interpreted as digestion. If it is a common preservation artifact of this site, it should be mentioned.

Corrosion of the enamel of NMMP 96 (M₃): we have completed the text to indicate the corrosion of the enamel of this specimen is the result of modern plant rootlet alteration, a commonly observed enamel alteration in the locality of PK2.

The drawing of the outline of the M1 looks a bit weird to me. Could it not have been a bit shifted lingually on the figure? The logic of this outlined M1 should be better explained in the text.

We have corrected the interpretive drawing of the holotype maxilla (position of the M1/ corrected by a slight buccal shift of the tooth outline).

Finally, regarding the CT aspects, I would like to have much more details regarding the acquisition parameters, with at least the source voltage and intensity, the filters used, the number of projection and the exposure time / frame averaging. In general, these parameters should always be provided when using μ CT / CT data in order to be able to judge about the quality of the scanning. There is a mistake in the method part: the maxilla and the mandible are mentioned, when the specimens scanned are the mandible and the M3.

We have provided the required methodological technical information about the CT scans (except for the maxilla for which only a surface scan of a cast was made). The mistake in the methods section noted by the reviewer 1 has been corrected.

Still considering μ CT data, it would be useful to have the data used for this paper publicly available, as it becomes more and more often the case. If not possible, clear statement explaining why it is not possible and under which condition the data would be accessible would be necessary. Nevertheless, increasing the number of publicly available data for such fossil specimens is the only way to really make possible in depth comparisons going beyond the surface morphology only. Making data public when using CT scanning would be really valuable for the whole profession. When slices cannot be made available for legal reasons, at least surface files should be provided especially as 3D pdfs have been apparently generated.

We have submitted as supplementary files the 3D pdfs of the holotype maxilla NMMP 93 and the mandible NMMP 95 that have been extracted from the CT-scans of the mandible and the cast of the holotype maxilla). These 3D pdfs will thus be made available to a large audience as required by reviewer 1.

To conclude, this is a nice paper that I recommend for publication, providing that additional illustrations are included to demonstrate the validity of several key characters presented by the authors, and I strongly encourage to go for publicly available μ CT data.

Reviewer #2 (Remarks to the Author):

This new maxilla and more fragmentary mandible from Myanmar are new and interesting fossils to consider for anthropoid evolution. They add a new combination of dental characters from what is known in Asia. They will be controversial as their interpreted status as a "crown" anthropoid versus a "stem" anthropoid will be debated. Making the connection from Asia to African anthropoids is the heart of the matter in terms of an evolutionary sequence and biogeographical origins. I think that these new fossils are deserving of a publication in Nature. They will certainly provide discussion and debate. I would also point out to the authors that there is postcranial evidence from Shanghuang that indicates a more advanced anthropoid existed there with the eosimiids although no dental material has been described.

We have compared *Aseanpithecus* with all available dental data about anthropoids from the Paleogene of China and cannot further compare if these new data are not published. Thus, no modification on the text has been made following the remarks of reviewer 2.

Reviewer #3 (Remarks to the Author):

Jaeger and colleagues report the discovery of two jaw fragments and an isolated tooth from the Late Middle Eocene of Myanmar. They attribute these to a new genus and species of anthropoid primate, and offer an interpretation of the significance of its morphology in the context of its age and geographical location. Their main conclusion is that this new species reveals something significant about the phylogenetic and biogeographical connections among the Paleogene anthropoids of Asia and Afro-Arabia. This conclusion is based on the perceived similarities that this taxon shares with later anthropoids from Afro-Arabia.

I see potential problems with three key premises in their argument: 1) that these specimens belong to one taxon, 2) that these specimens share derived features with later Afro-Arabian anthropoids not seen in Asian Paleogene anthropoids, and 3) that the perceived similarities have a potentially coherent interpretation given their distribution within Afro-Arabian Paleogene anthropoids. I explain these concerns below, and attach an image that compiles comparative illustrations of upper molars pertinent to the discussion from various publications. The authors have seen these, I know, but I include them so that they can see some of what I was looking at when making the observations that follow.

1) Attribution of the specimens. The new species is known from three specimens – an upper and lower jaw, and an isolated m3. These seem to match adequately in size, and all appear to belong to an anthropoid, so their association is understandable. However, there are other anthropoid taxa in the same beds, and the number of taxa has grown over the years. So it is not unrealistic to think that there is more than one previously undescribed species from this locality, rather than just one. After all, years ago, there were two primates there, then three, then four...

This is always a potential problem when establishing new taxa, and I only raise it here because the upper and lower jaws seem to resemble very different sets of taxa, as will be made clear below. This makes me worry that the two jaw fragments belong to different taxa.

The first point raised by reviewer 3 concerns the attribution of the lower jaw to the same taxon as the maxilla. We have based this attribution on three points: first, maxilla and mandible were discovered very close to each other (a few decimeters only), they match in size (as noted by reviewer 3) and the characters displayed by both the lower jaw and the maxilla are do not match any other fossil primate recovered since twenty years in this Formation. In addition, these two specimens share some morphological characters as the robust canines and anterior dentition, and a good occlusion between upper and lower premolars. Following the comments of reviewer 3 (in point 3), we have treated the maxilla and lower teeth of *Aseanpithecus* as separate OTUs. In order to treat the anthropoid taxa of the datamatrix homogeneously, we have decided to apply the same protocol to all anthropoids for which the association of upper and lower teeth is not firmly demonstrated (i.e., convincing evidence of lower and upper jaw association from a single individual). Considering that some taxa are also known from cranial and postcranial characters, these features have been grouped in a single OTU with those of upper jaws while lower jaw features have been treated as a second OTU.

The treatment of upper and lower jaw features in separated OTUs resulted in poorly-resolved and aberrant phylogenies (strict consensus with huge polytomy and anthropoid clade absent; Figure 1A below). In the majority-rule consensus (Figure 1B below), *Aseanpithecus* as well as most other anthropoids see their upper and lower jaw OTUs well separated in the obtained topology. This tree also often presents clades composed either of lower jaw OTUs or upper jaw OTUs. Phylogenies obtained with either only lower jaw OTUs (Figure 2 below) or upper jaw/skull/postcranial OTUs (Figure 3 below) are also very poorly-resolved and aberrant in several respects (e.g., tarsiids nested within anthropoids and Thai amphipithecid *Krabia* nested within platyrrhines in figure 2B, monophyly of oligopithecids and eosimiids lost and *Siamopithecus* nested within platyrrhines in figure 3B). The topologies obtained in these two analyses are also markedly different (e.g., eosimiids clade obtained in analysis 2).

These analyses indicate clearly that lower and upper teeth bear different phylogenetic signals and that current knowledge about the phylogeny of anthropoids results from the combination of both signals. Hence, the fact that the lower and upper teeth of *Aseanpithecus* present different affinities does not necessarily mean that this taxon is a chimera but represents instead a normal situation among anthropoids. In addition, the poorly-resolved and aberrant phylogenies obtained in these analyses precludes the separation of lower and upper teeth in distinct OTUs when the association of lower and upper jaw is most reasonable as it is the case for *Aseanpithecus*.

2) The apomorphies of *Aseanpithecus*. It is argued in this manuscript that *Aseanpithecus* exhibits a set of derived morphologies not seen in other Paleogene Asian anthropoids. In the upper molar(s) these are chiefly the reduced buccal cingula, parastyles, metastyles, paraconules and metaconules, enlarged trigon basin, and reduced concavity of the distal molar margin. I believe that these are either seen, or are easily produced by wearing the known molars of *Bahinia*, *Phileosimias*, and *Bugtipithecus*. The upper molars show fairly heavy wear, despite the apparent lack of exposed dentin. The buccal surface of the protocone is clearly very worn. This wear would, because of the geometry of the protocone, preprotocrista, and postprotocrista, have the effect of enlarging the basin. The morphology seen here could easily be achieved by wearing down the lingual side of a molar that started out with a more

restricted basin, like those of *Bahinia pondaungensis* or *Phileosimias*. That such wear was occurring in this dentition is clearly corroborated by the wear on the premolars.

Concerning tooth wear, we have reexamined carefully the original specimen (in National Museum, Nai Pyi Taw during August 2018) and we confirm that the M2/ of the holotype is not at all affected by tooth wear and that its present morphology is original. The pre- and postprotocrista are very slightly worn and, inside the trigone basin, some tiny enamel wrinkles can be recognized, even on our supplementary figure 1, which testify that the trigone basin was not worn at all. Reviewer 3 recognizes that there is no dentine pit which represents in itself an observation corroborating the absence of main cusps apical wear. The same is true for the premolars, which have acute cusp tips displaying no or very small dentine pits. Therefore we maintain that this maxilla displays its original tooth anatomy and that he has not been heavily worn or eroded.

The buccal portion of the molar crowns do lack the pronounced styles and shelf-like cingulum often seen in eosimiids, but these features are poorly developed in some taxa – like *Bahinia pondaungensis*, *Bugtipithecus*, and *Phileosimias*. These taxa also lack the pronounced concavity of the distal crown margin. At this point, it is worth noting that much of this could also be said of *Altiatlasius* from the Paleocene of Africa. The lack of any reference to this genus in the paper is a major shortcoming.

We have examined carefully and in great depth the critics formulated by this reviewer and subsequently added *Altiatlasius* in the compared taxa.

The upper premolars of *Aseanpithecus* are, in my view, not demonstrably different from those of *Bahinia*. The morphology of *Bahinia*'s premolars has been difficult to discern with certainty because the original illustration in Jaeger (1999) inaccurately reconstructs them. Comparison of more recent imagery of the specimen shows that the P₃ and P₄ are incomplete buccally and heavily worn lingually. This has several consequences. First, the relative buccolingual dimensions of the two teeth cannot be precisely determined. The P₃ seems narrower, but the incomplete nature of both teeth, and the potential buccal displacement of the P₃ crown, prevent us from falsifying the possibility that the two are very similar in breadth (as in *Aseanpithecus*). Second, the assertion that *Bahinia* lacks a P₃ protocone is difficult to confirm in the type specimen. This similarity to *Bahinia* combined with the near lack of direct evidence for the premolar morphology of *Phileosimias* and *Bugtipithecus* leaves open the possibility that there is little difference among them. I realize that the P₃ assigned to *Bahinia banyueae* has no protocone, but the generic assignment of that species could be debated. In sum, the known upper teeth of *Aseanpithecus* do not seem very different from those of several known Asian Paleogene anthropoids. It is entirely possible that it is closely related, if not congeneric, to one of them.

In the lower dentition, the main apomorphy is the relative size of the P₂ and P₃, with the P₂ being at least as large a crown (though single-rooted) as the P₃. This feature of the lower premolars is indeed unlike the size sequence in *Eosimias*, *Myanmarpithecus*, and *Pondaungia*, but *Bugtipithecus*, *Phileosimias* and *Siamopithecus* are not known from specimens that can demonstrate the size sequence of the premolars, and *Bahinia pondaungensis* is difficult to interpret with certainty (the known mandibular specimen has a P₃ that is damaged in such a way that it appears to be larger than P₄, which is surely an artifact). These taxa, in which the premolar size sequence is ambiguous, include taxa that I've argued resemble *Aseanpithecus* in the morphology of its upper dentition. This being the case, it is again possible that

Aseanpithecus (assuming that its upper and lower jaws belong together) is a close relative of something like *Phileosimias*, *Bugtipithecus*, or *Bahinia*.

We have also reexamined the teeth of the holotype of *Bahinia* at the National Museum in Nae Pyi Taw and confirm that the premolars and molars show no wear facets and that the P3/ displays no protocone. This tooth presents only a short buccolingual expansion and a length/breadth ratio of 77% to the contrary of the P3/ of *Aseanpithecus* which presents a more developed lingual lobe bearing a protocone and possesses a length/breadth ratio of only 66%. P3/ and P4/ outlines of *Bahinia* are well preserved and clearly indicate that the outline of these teeth are significantly different from those of *Aseanpithecus*. The upper molars of *Bahinia* display paraconule, distinct hypoparacrista and important parastyle and metastyle shelves that are absent on the much more derived molars of *Aseanpithecus*. Reviewer 3 mentions (obviously by mistake) the occurrence of a paraconule on *Aseanpithecus* M2/ but the arrow on his/her figure is pointing in fact to a tiny metaconule. *Bahinia* lower premolars are also very different from those of *Aseanpithecus* especially in the large P/3 which displays a well preserved outline, very distinct of that of *Aseanpithecus*, but similar to that of *Bahinia banyueae* from the lower Oligocene of China. The P/2 of *Bahinia* is also reduced in comparison to the large P/2 of *Aseanpithecus*. The lower premolar series of *Siamopithecus*, are well known (Chaimanee et al. 2000, C. R. Acad. Sci. Paris, 323, 235-241), with a very small single-rooted P/2, a larger unicuspid P/3 without metaconid and talonid basin and a larger P/4 with a metaconid and a small talonid basin. *Siamopithecus* lower premolars appear to be more primitive than those of *Bahinia* and *Aseanpithecus*. *Bugtipithecus* occupies sometimes a close position to *Aseanpithecus* in phylogenetic parsimonious analyses. However, its upper molars display numerous differences such as a strong hypocone with a distinct prehypocrista, distinct para- and metaconules and strong parastyles, and the somehow twisted trigonid crests.

The primate diversity of Pondaung Formation has certainly increased during the last 20 years of excavations, but always within the same register: amphipithecids, eosimiiforms and sivaladapids. We demonstrate in the text that while the new fossils cannot be referred to any of these groups, it is absolutely not surprising that *Aseanpithecus* shares primitive characters with the eosimiid *Bahinia* since eosimiids are considered as the basalmost group of all anthropoids! Considering that these Asian taxa arose from a radiation of stem anthropoid primates which obviously started long before the radiation of African anthropoids, it is normal to find resemblances between their dentitions. Nevertheless, *Aseanpithecus* displays derived characters that were yet not reported from any Asian Eocene anthropoid primate. To summarize, we are confident that *Aseanpithecus* teeth characters are original, and unlike any of the other Asian Eocene anthropoid primates and characterizes a new clade.

3) Distribution of *Aseanpithecus*'s apparent apomorphies among Afro-Arabian anthropoids.

This last issue may seem the most nebulous of the three, but in fact it may be the most problematic because it threatens the validity of their argument, while the first two problems concern only the veracity of individual premises. Although, as argued above, some of the purported apomorphies of *Aseanpithecus* are found in other Asian taxa, some have so-far only been documented in Afro-Arabian fossils. However, their distribution in Afro-Arabian taxa is

odd. The upper molar features are most closely matched by the stem catarrhine *Catopithecus* (in which the premolars are profoundly different), while the lower premolar traits are found in proteopithecids (in which the upper molars are completely different). So the lower teeth resemble proteopithecids (stem anthropoids), and the upper molar (but not premolars) resembles a stem catarrhine – and one that is typically considered more derived than *Oligopithecus*. Parenthetically, some of these molar traits are also present in the Paleocene *Altiatlasius*, which complicates their polarity. This pattern of distribution could support a premise like “*Aseanpithecus* exhibits features not found in other Asian Paleogene anthropoids, but found in various combinations in later Afro-Arabian Paleogene anthropoids”. As explained above, I’m not sure that even this is accurate because some of these traits are found in Asian taxa like *Phileosimias* and even in the *earlier* African *Altiatlasius*. So even this vague premise is generous. But my present point is that because these traits do not align *Aseanpithecus* with any particular Afro-Arabian clade, there is no particular paleobiogeographical scenario to which this argument can lead. It is tantalizing, and the specimens may well end up being a meaningful part of the solution, but at present they offer only an intriguing clue. They are suggestive of something, but what? I wouldn’t argue that such riddles can only be explored or solved by using massive cladistic data sets, but in such complicated situations, it is a good starting point. The authors have conducted such an analysis, and the unstable position of *Aseanpithecus* under various assumption sets reflects the ambiguity just described, but it also clearly shows its propensity for clustering with *Bugtipithecus*.

We have extended in the text the comparisons with *Bugtipithecus* and listed the characters that differentiate basically this taxon from *Aseanpithecus*.

If one tries (as does the parsimony algorithm) to hypothesize a way in which this new species connects more primitive Asian Paleogene anthropoids with later and more derived Afro-Arabian ones, one might imagine a dispersal of an *Aseanpithecus*-like group into Africa, where it is somehow involved in the evolution of oligopithecids or proteopithecids. But where do they fit? Are they somewhere between proteopithecids and *Catopithecus*? Do they show us the initial changes toward the catarrhine lineage? If this is the case, why does the upper molar resemble *Catopithecus* instead of *Oligopithecus*? And why would it exhibit a large P₂, when these are absent in oligopithecines? Did the lineage enlarge them before losing them? Or is it a stem anthropoid like a proteopithecid. If so, its similarities to *Catopithecus* are all convergent, and its only potential synapomorphy with anything in Africa is the large P₂ – which we can’t be sure is not present in some Asian taxa (or even *Altiatlasius*).

Even if I set aside my concerns about the composition of the hypodigm, and the overstated differences between *Aseanpithecus* and all known Asian taxa, and concede that the upper teeth look just like *Catopithecus*, and the lower teeth like proteopithecids, it still leaves us with no way of interpreting these similarities without one set being convergent and irrelevant. Honestly, I’d sooner believe that these fossils document two taxa - an early oligopithecid, *and* an early proteopithecid. I know it would be a more sensational claim, but it would follow more clearly from the available evidence. In fact, I’d encourage the authors to uncouple the upper and lower dentitions of *Aseanpithecus* and try them experimentally as separate OTUs. I’d be curious to see where the analysis puts them. Lastly, I want to make clear that it is entirely possible that the authors’ conclusions are right. I am pointing out what I see as problems with their argument. If these weaknesses cannot be remedied, then their argument is the worse for it, but hopefully my criticisms can improve their argument. As always, I am skeptical of the argument, not of the authors.

Concerning the third point raised by reviewer 3, which concerns the meaning of the African anthropoid characters present in *Aseanpithecus*, we would like to remember that in Pondaung Formation, only stem anthropoids have been recorded. *Bahinia* is so primitive that it is recognized as a member of the basalmost group of anthropoids, the Eosimiidae. Amphipithecids are also stem anthropoids but show a highly specialized dento-gnathic morphology likely representing adaptations to the consumption of hard items. The discovery of an anthropoid with more derived dental characters and with some complex characters shared with fossil and extant African and South American crown anthropoids indicate that during the early radiation of anthropoids in Asia, some more derived group differentiated, which had the potential to generate combination of dental characters which may characterize African crown anthropoids. We do not suggest that *Aseanpithecus* is the direct ancestor of these African crown anthropoids, but rather consider it as a potential sister group to the anthropoids which colonized Africa from Asia during the Middle Eocene. This scenario can be only completed by the discovery of more complete representatives of these earlier immigrants in Africa and by new representatives of this clade in Asia.

Figure 1. Strict (1A) and majority-rule (1B) consensus obtained with lower jaw and upper jaw/cranial/postcranial characters treated as separate OTUs. 2762993 most parsimonious trees obtained (length

Figure 2. Strict (2A) and majority-rule (2B) consensus obtained with lower jaw OTUs only. 450 most-parsimonious trees of 1231 steps obtained.

Figure 3. Strict (3A) and majority-rule (3B) consensus obtained with upper jaw/cranial/postcranial OTUs only. 13366 most-parsimonious trees of 1177 steps obtained.

Reviewers' Comments:

Reviewer #1:

Remarks to the Author:

The reviewed manuscript and rebuttal letter covers most of the remarks I made during my first review. All in all, it is fine for me for publication, except for few small items linked to the description and to the 3D pdfs :

- vertical and unfused symphysis: I agree that the symphysis can be considered as vertical, but, as nothing from the symphyseal plan is preserved, I do not see on which basis the unfused character presented in the description can be obtained. It looks like an inference based on a most probable state of the character, but not like an observation coming from the specimen. I would then suggest to change the text into "most probably unfused symphysis" or something equivalent.

- The authors present the fact that the different fossils are coming from a very restricted area, that the maxilla and mandible are of similar size, similar wear level, and do present quite good occlusion of the premolars. I think that the authors should then discuss the possibility that these two (and possibly 3) specimens could be from a single individual, or to explain what would be the reasons to exclude (or to support) this hypothesis. In the same way, as the authors are now providing 3D pdfs, I would strongly encourage them to make a third one with a composite reconstruction of the specimen in order to better illustrate the dental arcade shapes, the occlusion pattern, and all the related aspects in addition to the 2D views of symmetric samples. It would also be useful regarding the comments of the two other reviewers about the association of these specimens within a single taxon.

- Finally, two small typos, there is a sentence line 312 that is strikethrough, I suppose that it should be completely removed. Line 422 I guess that it should be "as" not "are".

To conclude, I recommend this paper for publication.

Reviewer #3:

Remarks to the Author:

In my view, which I had hoped would be one voice out of three reviewers equally capable of commenting on the interpretation of these Paleogene primates, the authors are making more of these specimens that can be responsibly made.

Their comparisons of the new specimens to known taxa are riddled with claims of similarity and difference that I believe are inaccurate and misleading. Their response to my review indicates that they cannot tell when a specimen is damaged or worn, or refuse to acknowledge it (this pertains as much to other taxa with which they are comparing the new material as to the new fossils themselves). They claim that I have mistakenly identified a paraconule on their M2, and that the cuspule I indicated with an arrow on my review figure is actually a small metaconule. I'm glad to see that they acknowledge the existence of this cuspule, but regret to inform them that it is, in fact, the paraconule because the image in question is one that I reversed from left to right in order to make it directly comparable to the other teeth in the figure. So it seems that they can't tell which way this "unworn" tooth is oriented by looking at it, which concerns me.

Regarding the wear on this molar, I will say that the 3D pdf now provided (thank you, Paul) shows that it is not as worn as I had feared, but the excellent photograph clearly shows a flattened wear surface with linear striations – these are not enamel wrinkles. I'm also a little surprised at their response to this point that they "reexamined" the original specimen in August. That was long before my review.

They insist that the crown outlines of P3 and P4 in Bahinia are well preserved. This nonsense can only be supported if one looks only at the poorly drawn representations of these teeth. The actual fossils are broken and missing buccal portions of the crowns.

The concatenated phylogenetic experiments that they have added do not address my concern about the differences between the upper and lower teeth. They've missed my point in this regard.

I don't see any benefit in further countering their response to my review, because they have made it clear that they will not accept the criticisms I've made (except when they accidentally do so by mistakenly agreeing with my identification of a feature on a reversed image).

I do not think that this is worthy of appearing in anything with Nature in the title. It would need major revision to appear in The Journal of Human Evolution, and there it would at least have the benefit of receiving reviews from two or three people who know something about the relevant fossils.

Reviewer #4:

Remarks to the Author:

Generally I think this is a well-written and clearly argued paper. Although I'm not necessarily convinced by the authors' conclusion about the systematic importance of this new taxon, with a few exceptions (see below) the argument is generally well made, and deserves to be published so that this interpretation can be debated in an open forum. The alternative (that the similarities between this new taxon and the African crown anthropoids are convergences) isn't really testable without further data, so even though I'm not necessarily convinced, I think they've made the best argument that can be made for the meaning of the morphology of the next taxon.

With one exception, I find the interpretations of the anatomy are sound. The one exception has to do with the angle of the mandibular symphysis. I carefully reviewed the additional figure that the authors provided as part of their response to the reviewers (which should be included in the Supplementary Data). Although I'm open to the possibility that they've identified part of the distal edge of the planum alveolare (as indicated with arrows in that figure), I don't see any reason why the point they indicated must be the distal termination of the symphysis—it just looks like a broken edge to me. Also, I'm not convinced that, even if they've gotten all of these points correct, they necessarily know from that information what the front of the symphysis would look like. I think it would be better to focus on what they can say, which is that the incisors clearly weren't strongly horizontally inclined. But from my perspective, their degree of vertical inclination at the front of the jaw is an open question.

Ironically, something that would help them make their case is distribution of a version of the CT data that could be 3D printed, since it can be much easier to assess this type of question when you can experience the morphology in three dimensions. I was very disappointed that they responded to the reviewer's comment about distributing 3D data by indicating that they would make a 3D pdf available. The ethical standard in Paleontology is that, once a specimen is published, it should be made available for study by other workers. To my mind, that standard should extend to distribution of the actual CT data (the original tiffs), since if an argument is being made on the basis of those data, other researchers should be able to review them. I encourage the authors to rethink their decision on this point.

Finally, before I get to my detailed comments on specific parts of the paper, I would like to address some of the criticisms made by Reviewer 3 from this paper's previous submission, since I'm guessing I was added to the paper as a reviewer because of the harshness of that review. That Reviewer questioned the association between the upper and lower teeth. Although at one level that is a fair point, the argument as outlined in the response to the reviewers is as well made as it ever is for association between upper and lower teeth (as noted below, some of those points need to be put into

the paper itself). I think it's possible that the authors are wrong, but the same point could be made about a majority of fossil anthropoid taxa. So it's honestly not a very relevant point. Second, the reviewer makes a rather peculiar argument about how the effect of wear could be making this taxa appear to be distinct when it really is not. I don't understand where the reviewer is coming from at all—it is very clear from the photograph that the upper molars are little worn. They retain very fine enamel features, which would be worn off very quickly. The reviewer's argument frankly seems like an enormous stretch to try and undermine the distinctness of this taxa, with little coherent basis. Finally, the reviewer points out that there are crossing synapomorphies with respect to the upper and lower teeth (i.e., that the lower teeth look more like proteopithecids and the uppers like Catopithecus). Okay, yes, that's true. But that doesn't imply that all of the African anthropoid-like traits are homoplastic. The authors have provided a fair portrayal of the potential meaning of these crossing synapomorphies by including results from a range of phylogenetic analyses.

I do have some minor (and one not so minor) points that I think the authors should address before publication in order to strengthen the paper:

p. 5: generally the diagnosis is well written. However, there are a few points in which the authors have used terminology that implies an evolutionary trajectory, which should be avoided in diagnoses because it makes them ambiguous. Specifically: line 92: "reduced" is ambiguous—reduced relative to what?; line 95: I'm not sure what "more derived P3" means—if they mean in the ways listed in the rest of that sentence, then it would be preferable to remove "more derived"; line 96: "reduced" is again ambiguous; "small" or "tiny" would be preferable; line 97: again, "reduced" is ambiguous. Use a descriptive term instead; line 104: specify what is meant by "premolar size proportions"; line 107: replace "less derived P3-P4 morphology" with a description of what that means

p. 7, line 151: the nasal opening is "rather large" compared to what? Clarify

p. 9, line 200: I don't know what "high elevation of the dentary" means. High relative to what? What part is "high"? Clarify.

p. 9, line 201: "presents an adequate occlusion with the upper premolars of the maxilla" is pretty weak. Please explain more clearly what you did to assess this, and how the occlusion was deemed to be "adequate". This is better explained in the response to the reviewers, so those points should be added to the paper (as noted above).

p. 9, line 204: replace "his" with "its"

p. 9 line 207: as noted above, I am not convinced that the evidence supports the conclusion about the "short and strongly inclined symphyseal area". Partially wrt "short" the whole front is missing—how could you possibly know?

p. 12, lines 275-276: "Most of the resemblances between these two taxa correspond to shared primitive characters." Based on what model of primitiveness? You cannot just assert this without a comparative context. Need to clarify or cut.

p. 16, line 366: We don't use "important" in this way in English. Replace with "significant" or "large"

p. 18 lines 416-423: This section weakens the paper. The first sentence "By just recombining..." actually makes no sense—it's like saying "if I invent a taxon it will have these relationships". And the rest of the argument is based on an unsupported view of how dental changes occur genetically. If there is anything that we've learned from our burgeoning understanding of how morphological change

occurs genetically, it is that our intuitive sense of it is often wrong, with much more coordinated change possible. So to say "Homoplasy can affect some simple characters but not a set of complex characters are [sic] seen in Aseanpithecus..." is just not something that can be asserted without support. This argument is poorly made and not necessary—I would recommend just cutting it.

Mary T. Silcox, PhD

Reviewers' comments:

Reviewer #1 (Remarks to the Author):

The reviewed manuscript and rebuttal letter covers most of the remarks I made during my first review. All in all, it is fine for me for publication, except for few small items linked to the description and to the 3D pdfs :

- vertical and unfused symphysis: I agree that the symphysis can be considered as vertical, but, as nothing from the symphyseal plan is preserved, I do not see on which basis the unfused character presented in the description can be obtained. It looks like an inference based on a most probable state of the character, but not like an observation coming from the specimen. I would then suggest to change the text into "most probably unfused symphysis" or something equivalent.

Done 1-Line 88: We used "most probably unfused symphysis" as suggested by reviewer 1

- The authors present the fact that the different fossils are coming from a very restricted area, that the maxilla and mandible are of similar size, similar wear level, and do present quite good occlusion of the premolars. I think that the authors should then discuss the possibility that these two (and possibly 3) specimens could be from a single individual, or to explain what would be the reasons to exclude (or to support) this hypothesis.

Done 2: Line 239: We added some text to explain why we don't support that these fossils belong to the same individual.

In the same way, as the authors are now providing 3D pdfs, I would strongly encourage them to make a third one with a composite reconstruction of the specimen in order to better illustrate the dental arcade shapes, the occlusion pattern, and all the related aspects in addition to the 2D views of symmetric samples. It would also be useful regarding the comments of the two other reviewers about the association of these specimens within a single taxon.

We have provided a composite 3D pdf produced with symmetrized maxilla and mandible as required. This 3D pdf has been deposited in figshare together with other 3D pdfs (isolated jaw and maxilla). See file 'Aseanpithecus_related_files_figshare_links.txt' for links to these files.

- Finally, two small typos, there is a sentence line 312 that is strikethrough, I suppose that it should be completely removed.

Done 4: Line 312: They have been removed

Line 422 I guess that it should be "as" not "are".

Done 5: Line 422. The whole paragraph has been deleted on recommendation of reviewer 4.

To conclude, I recommend this paper for publication.

Reviewer #3 (Remarks to the Author):

In my view, which I had hoped would be one voice out of three reviewers equally capable of commenting on the interpretation of these Paleogene primates, the authors are making more of these specimens that can be responsibly made.

Their comparisons of the new specimens to known taxa are riddled with claims of similarity and difference that I believe are inaccurate and misleading. Their response to my review indicates that they cannot tell when a specimen is damaged or worn, or refuse to acknowledge it (this pertains as much to other taxa with which they are comparing the new material as to the new fossils themselves).

Concerning the new described fossil, its upper molars are very slightly worn and they have “retained fine enamel features which would be worn off very quickly” according to reviewer 4.

They claim that I have mistakenly identified a paraconule on their M2, and that the cuspule I indicated with an arrow on my review figure is actually a small metaconule. I’m glad to see that they acknowledge the existence of this cuspule, but regret to inform them that it is, in fact, the paraconule because the image in question is one that I reversed from left to right in order to make it directly comparable to the other teeth in the figure. So it seems that they can’t tell which way this “unworn” tooth is oriented by looking at it, which concerns me.

We have clearly indicated in the original main text Line 183 and 184 that “no paraconule is present” but than “a tiny swelling corresponding to a metaconule is visible”. The paraconule was not erased by wear as suggested by reviewer 3 because this tooth displays very little wear and, therefore, its absence indicate that it was originally not present as on *Catopithecus*. The same is true for M3/. The preparaconule crista was also absent and the preprotocrista ends directly in contact with the parastyle (supplementary fig. 1).

Regarding the wear on this molar, I will say that the 3D pdf now provided (thank you, Paul) shows that it is not as worn as I had feared, but the excellent photograph clearly shows a flattened wear surface with linear striations – these are not enamel wrinkles. I’m also a little surprised at their response to this point that they “reexamined” the original specimen in August. That was long before my review. They insist that the crown outlines of P3 and P4 in *Bahinia* are well preserved. This nonsense can only be supported if one looks only at the poorly drawn representations of these teeth. The actual fossils are broken and missing buccal portions of the crowns.

We recognize that the available illustrations of *Bahinia* premolar are not of outstanding quality (especially in *Primate Fossil record*, P. 139, fig.9.4H). But the original publication in *Science* (1999) gives drawings of two maxilla of the same individual. Both display complete outline of the P3/ from which it appears that there is no alteration of their buccal portion nor of their lingual wall (especially NMMP 15) and that no protocone was present, unlike *Aseanpithecus*. The left P4/ (NMMP 14) has also an undisturbed outline. Our comparisons have been made on original fossils, not on casts or pictures.

The concatenated phylogenetic experiments that they have added do not address my concern about the differences between the upper and lower teeth. They’ve missed my point in this regard.

We have done these concatenated phylogenetic experiments on request of reviewer 3. They produced no valuable results. Finally, we don’t understand why he claims that we have missed his point.

I don’t see any benefit in further countering their response to my review, because they have made it clear that they will not accept the criticisms I’ve made (except when they accidentally do so by mistakenly agreeing with my identification of a feature on a reversed image). I do not think that this is worthy of appearing in anything with *Nature* in the title. It would need major revision to appear in

The Journal of Human Evolution, and there it would at least have the benefit of receiving reviews from two or three people who know something about the relevant fossils. **No comments!**

Reviewer #4 (Remarks to the Author):

Generally I think this is a well-written and clearly argued paper. Although I'm not necessarily convinced by the authors' conclusion about the systematic importance of this new taxon, with a few exceptions (see below) the argument is generally well made, and deserves to be published so that this interpretation can be debated in an open forum. The alternative (that the similarities between this new taxon and the African crown anthropoids are convergences) isn't really testable without further data, so even though I'm not necessarily convinced, I think they've made the best argument that can be made for the meaning of the morphology of the next taxon.

With one exception, I find the interpretations of the anatomy are sound. The one exception has to do with the angle of the mandibular symphysis. I carefully reviewed the additional figure that the authors provided as part of their response to the reviewers (which should be included in the Supplementary Data). Although I'm open to the possibility that they've identified part of the distal edge of the planum alveolare (as indicated with arrows in that figure), I don't see any reason why the point they indicated must be the distal termination of the symphysis—it just looks like a broken edge to me. Also, I'm not convinced that, even if they've gotten all of these points correct, they necessarily know from that information what the front of the symphysis would look like. I think it would be better to focus on what they can say, which is that the incisors clearly weren't strongly horizontally inclined. But from my perspective, their degree of vertical inclination at the front of the jaw is an open question.

Ironically, something that would help them make their case is distribution of a version of the CT data that could be 3D printed, since it can be much easier to assess this type of question when you can experience the morphology in three dimensions. I was very disappointed that they responded to the reviewer's comment about distributing 3D data by indicating that they would make a 3D pdf available. The ethical standard in Paleontology is that, once a specimen is published, it should be made available for study by other workers. To my mind, that standard should extend to distribution of the actual CT data (the original tiffs), since if an argument is being made on the basis of those data, other researchers should be able to review them. I **encourage the authors to rethink their decision on this point.**

We have decided following these remarks to give access to the full CT data of the lower jaw. These data have been deposited in figshare together with available 3D printable surface files (.stl) and 3D pdfs of *Aseanpithecus* fossils. See file 'Aseanpithecus_related_files_figshare_links.txt' for links to these files.

Finally, before I get to my detailed comments on specific parts of the paper, I would like to address some of the criticisms made by Reviewer 3 from this paper's previous submission, since I'm guessing I was added to the paper as a reviewer because of the harshness of that review. That Reviewer questioned the association between the upper and lower teeth. Although at one level that is a fair point, the argument as outlined in the response to the reviewers is as well made as it ever is for association between upper and lower teeth (as noted below, some of those points need to be put into the paper itself). I think it's possible that the authors are wrong, but the same point could be

made about a majority of fossil anthropoid taxa. So it's honestly not a very relevant point. Second, the reviewer makes a rather peculiar argument about how the effect of wear could be making this taxa appear to be distinct when it really is not. I don't understand where the reviewer is coming from at all—it is very clear from the photograph that the upper molars are little worn. They retain very fine enamel features, which would be worn off very quickly. The reviewer's argument frankly seems like an enormous stretch to try and undermine the distinctness of this taxa, with little coherent basis. Finally, the reviewer points out that there are crossing synapomorphies with respect to the upper and lower teeth (i.e., that the lower teeth look more like proteopithecids and the uppers like *Catopithecus*). Okay, yes, that's true. But that doesn't imply that all of the African anthropoid-like traits are homoplastic. The authors have provided a fair portrayal of the potential meaning of these crossing synapomorphies by including results from a range of phylogenetic analyses.

I do have some minor (and one not so minor) points that I think the authors should address before publication in order to strengthen the paper:

p. 5: generally the diagnosis is well written. However, there are a few points in which the authors have used terminology that implies an evolutionary trajectory, which should be avoided in diagnoses because it makes them ambiguous. Specifically: line 92: "reduced" is ambiguous—reduced relative to what?;

Done 6: We have suppressed "and reduced" for the description of M/3 trigonid.

line 95: I'm not sure what "more derived P3" means—if they mean in the ways listed in the rest of that sentence, then it would be preferable to remove "more derived";

Done 7: We have suppressed "more derived".

line 96: "reduced" is again ambiguous; "small" or "tiny" would be preferable;

Done 8: We have replaced it by "small"

line 97: again, "reduced" is ambiguous. Use a descriptive term instead;

Done 9: We have replaced it by "less distinct (hypoparacrista).

line 104: specify what is meant by "premolar size proportions"

Done 10: We have replaced size by "premolar length/width proportions ;

line 107: replace "less derived P3-P4 morphology" with a description of what that means

Done 11: We have suppressed that sentence and replaced it by "P³-P⁴ more buccolingually extended, P³ with a more mesially located paracone and a sharper and longer postparacrista, with smaller protocone and no lingual cingulum".

p. 7, line 151: the nasal opening is "rather large" compared to what? Clarify

Done 12: We have suppressed the whole sentence concerning nasal opening.

p. 9, line 200: I don't know what "high elevation of the dentary" means. High relative to what? What part is "high"? Clarify.

Done 13: We have replaced it by "horizontal ramus of the mandible".

p. 9, line 201: “presents an adequate occlusion with the upper premolars of the maxilla” is pretty weak. Please explain more clearly what you did to assess this, and how the occlusion was deemed to be “adequate”. This is better explained in the response to the reviewers, so those points should be added to the paper (as noted above).

Done 14: We have added here in the new text the arguments given previously to reviewers to support the association between maxilla and lower jaw. We have suppressed the sentence containing the reference to an “adequate occlusion”.

p. 9, line 204: replace “his” with “its” **Done 15.**

p. 9 line 207: as noted above, I am not convinced that the evidence supports the conclusion about the “short and strongly inclined symphyseal area”. Partially wrt “short” the whole front is missing—how could you possibly know?

Done 16: We have modified this paragraph in adding a supplementary Fig. 1 showing two detailed pictures of the symphysis, added “possible” for the “lowermost extremity”, suppressed “suggesting a rather short and strongly inclined symphyseal area” and added instead “and combined with the angle made by the planum alveolare it suggest that incisors roots were not strongly horizontally inclined.”

p. 12, lines 275-276: “Most of the resemblances between these two taxa correspond to shared primitive characters.” Based on what model of primitiveness? You cannot just assert this without a comparative context. Need to clarify or cut.

Done 17: We have suppressed this sentence.

p. 16, line 366: We don’t use “important” in this way in English. Replace with “significant” or “large”

Done 18: We have replaced it by “significant”.

p. 18 lines 416-423: This section weakens the paper. The first sentence “By just recombining...” actually makes no sense—it’s like saying “if I invent a taxon it will have these relationships”. And the rest of the argument is based on an unsupported view of how dental changes occur genetically. If there is anything that we’ve learned from our burgeoning understanding of how morphological change occurs genetically, it is that our intuitive sense of it is often wrong, with much more coordinated change possible. So to say “Homoplasy can affect some simple characters but not a set of complex characters are [sic] seen in Aseanpithecus...” is just not something that can be asserted without support. This argument is poorly made and not necessary—I would recommend just cutting it.

Done 18: We have deleted the whole paragraph.

Mary T. Silcox, PhD

Reviewers' Comments:

Reviewer #4:

Remarks to the Author:

I have reviewed the authors' response to the previous reviews, and read through the new version of the manuscript. In my opinion, they have made more than adequate responses to the reviewers' comments. In particular, I am delighted by their decision to make the original tiff files of the CT scan available. In my estimation, the paper is now ready for publication.